# pH-gated nanoparticles selectively regulate lysosomal function of tumour-associated macrophages for cancer immunotherapy

Mingmei Tang[1,2], Binlong Chen [2], Heming Xia[2], Meijie Pan[2], Ruiyang Zhao[2], Jiayi Zhou[2], Qingqing Yin[2], Fangjie Wan[2], Yue Yan[2], Chuanxun Fu[2], Lijun Zhong[3], Qiang Zhang[1,2] & Yiguang Wang [1,2,4] ✉

Tumour-associated macrophages (TAMs), as one of the most abundant tumour-infiltrating immune cells, play a pivotal role in tumour antigen clearance and immune suppression. M2-like TAMs present a heightened lysosomal acidity and protease activity, limiting an effective antigen cross-presentation. How to selectively reprogram M2-like TAMs to reinvigorate anti-tumour immune responses is challenging. Here, we report a pH-gated nanoadjuvant (PGN) that selectively targets the lysosomes of M2-like TAMs in tumours rather than the corresponding organelles from macrophages in healthy tissues. Enabled by the PGN nanotechnology, M2-like TAMs are specifically switched to a M1-like phenotype with attenuated lysosomal acidity and cathepsin activity for improved antigen cross-presentation, thus eliciting adaptive immune response and sustained tumour regression in tumour-bearing female mice. Our findings provide insights into how to specifically regulate lysosomal function of TAMs for efficient cancer immunotherapy.

The tumour microenvironment is profoundly immunosuppressive and hampers the response to immunotherapy treatment such as checkpoint blockade and vaccination[1]. As the most abundant immune cells in the tumour microenvironment, tumour-associated macrophages (TAMs) predominantly display a tumour-promoting M2-like phenotype that plays a crucial role in tumour progression, metastasis, immune evasion, and resistance to immunotherapy[2]. Increasing evidence demonstrates that a high abundance of TAMs is associated with unfavourable clinical outcomes in a wide range of cancer types[3,4]. TAMs display the capacity to suppress cytotoxic T cell function directly and indirectly by the following factors, including (i) production of inhibitory cytokines (such as IL-10)[5], (ii) intervention of antigen presentation[6] and (iii) inhibition of stimulatory populations (e.g. dendritic cells, DCs) and recruitment of immunosuppressive populations (e.g. regulatory T cells)[7]. Currently, a variety of anti-TAMs strategies have been evaluated in preclinical research and different stages of

clinical trials[8–10]. However, it remains a paramount challenge to specifically manipulate TAMs to promote anti-tumour immune response while sparing macrophages in healthy tissues.

Mature DCs are well-recognised as professional antigen-presenting cells with a dial-down lysosomal function, such as increased compartmental pH and low levels of lysosomal enzymes to maximise the generation of suitable peptides for antigen presentation[11,12]. TAMs in the M2-like phenotype present an upregulated lysosomal function that easily causes antigen degradation and is immunologically silent[8,13]. Conversely, pro-inflammatory M1-like macrophages with reduced lysosomal function also have the potential for antigen cross-presentation that activates CD8[+] T cells for effective tumour elimination[14]. Moreover, cross-presentation by anti-inflammatory phenotypes might lead to immune tolerance against self-proteins, similar to immature DCs behaviour[15]. Thus, we hypothesised that selective regulation of the lysosomal function of M2-like

[1]State Key Laboratory of Natural and Biomimetic Drugs, School of Pharmaceutical Sciences, Peking University, Beijing, China. [2]Beijing Key Laboratory of Molecular Pharmaceutics and New Drug Delivery System, School of Pharmaceutical Sciences, Peking University, Beijing, China. [3]Center of Medical and Health Analysis, Peking University Health Science Center, Beijing, China. [4]Chemical Biology Center, Peking University, Beijing, China. ✉ e-mail: yiguang.wang@pku.edu.cn

TAMs would switch their protumoural phenotype to tumouricidal M1-like ones, thereby fine-tuning their proteolysis for efficient antigen presentation to cytotoxic T lymphocytes.

Herein, we show a library of pH-gated nanoparticles (PGNs) that distinguish the acidic lysosomal milieu ($pH_L$) of different immune cells, allowing the specific targeting and reprogramming of M2-like macrophages in the tumour microenvironment instead of the immunocytes in the healthy tissues. PGN library consists of 11 nanoparticles with a pH transition ($pH_t$) from 4.5 to 5.5 and 0.1 pH increment, covering the lysosomal pH range of various cell types. We perform a screening and identify $PGN_{4.9}$ ($pH_t$ = 4.9) with an optimal pH-gated capacity for specific reporting of highly lysosomal acidity of M2-like macrophages ($pH_L$ ~4.4) rather than that of other cells ($pH_L$ ~5.2, Fig. 1). Furthermore, we design a $PGN_{4.9}$ nanoadjuvant with AND-gated capacity that converts two tandem signals, including highly lysosomal acidity and cathepsin activity, into efficient release of toll-like receptors 7/8 (TLR7/8) agonist (imidazoquinoline, IMDQ[16,17]) for selectively resetting M2-like phenotype to M1-like ones. $PGN_{4.9}$ nanoadjuvant efficiently tunes down lysosome acidification and proteolysis of M2-like TAMs, thereby potentiates antigen cross-presentation and activates $CD8^+$ T cell function for robust cancer immunotherapy. In contrast, $PGN_{4.9}$ nanoadjuvant keeps inert in other cell types and circumvents acute systemic toxicity. Combination immunotherapy with $PGN_{4.9}$ nanoadjuvant plus chemotherapy or checkpoint inhibitors induces sustained tumour regression and prolonged survival. Our work provides a nanoplatform to specifically reprogramme M2-like TAMs towards M1 phenotype for macrophage-based tumour immunotherapy.

## Results

### Design and characterisation of PGNs library

We firstly prepared M0-, M1- and M2-like macrophages with distinct functions by stimulating bone marrow-derived macrophages (BMDMs) with macrophage colony-stimulating factor (m-CSF), lipopolysaccharides (LPS) plus interferon-γ (IFN-γ), and interleukin 4 (IL-4) in vitro, respectively[18] (Supplementary Fig. 1). Then, a cocktail of Oregon Green-dextran and Rhodamine B-dextran was utilised to determine the lysosomal pH ($pH_L$) of various living cell types, including immune cells, tumour cells, and normal cells using ratiometric pH quantification protocol[19] (Supplementary Figs. 2 and 3). The results indicated that the lysosomes of M2-like BMDMs ($pH_L$ ~4.43) were hyper-acidified as compared with other cell types ($pH_L$ > 5.20, Fig. 2a).

We then set out to engineer PGNs with the capacity to differentiate the subtle lysosomal pH deviation between M2-like BMDMs and other cell types. Using our developed ultra-pH-sensitive nanotechnology[20–23], we synthesised a library of PGNs self-assembled from the amphiphilic copolymers (PEG-$b$-P($R_1$-$r$-$R_2$)), where poly(ethylene glycol) methyl ether (mPEG) chain works as a hydrophilic block and P($R_1$-$r$-$R_2$) serves as an ionisable hydrophobic block (Supplementary Fig. 4). The p$K_a$ of synthetic copolymers covered the pH range of lysosomal lumen ($pH_L$ ~4.5–5.5) for various cell types with pH increments of 0.1 as determined by pH titration method (Supplementary Fig. 5). Furthermore, the $pH_t$ of PGNs was determined to be within a range from 4.46 to 5.47 with sharp response ($\Delta pH_{ON/OFF}$ ~ 0.2–0.3, Fig. 2b) using fluorescence analysis procedure[21]. To evaluate the pH-gating capacity of PGNs in living cells, we successfully constructed binary ratiometric nanoreporters,

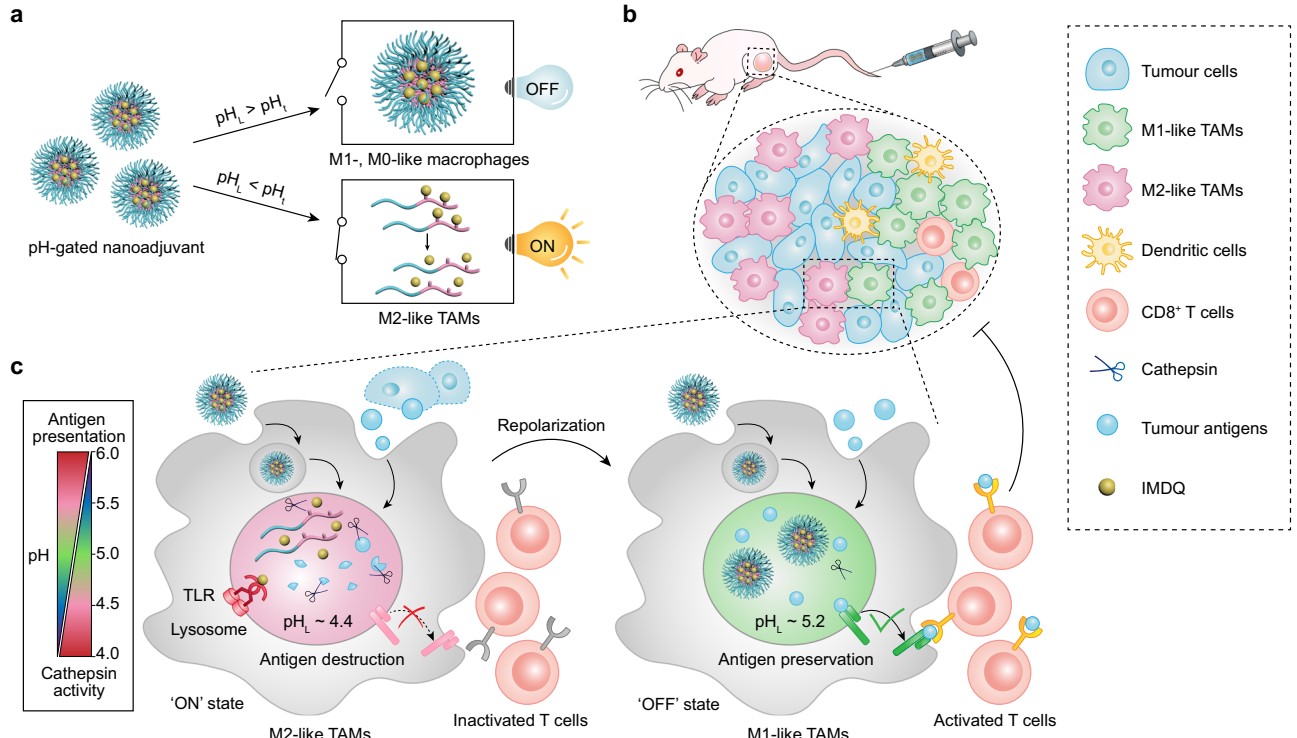

**Fig. 1 | Design and mechanism of $PGN_{4.9}$ nanoadjuvant for reprogramming M2-like TAMs into M1 phenotype via pH-gated regulation of lysosomal function.** **a** Development of pH-gated nanoadjuvant for activatable cancer immunotherapy in response to lysosomal acidity of M2-like TAMs. **b** Engineered nanoadjuvant can selectively target M2-like TAMs in tumour tissues after intravenous injection. **c** Upon endocytosis, nanoadjuvant is dissociated into polymer-GFLG-IMDQ conjugates unimer in the highly acidic lysosomes ($pH_L$ ~4.4) of M2-like TAMs, whereas keeps integrity in moderate acidic lysosomes ($pH_L$ ~5.2) of other cells, including M0-, M1-like macrophages, tumour cells, and normal cells. The highly expressed

lysosome proteases in M2-like TAMs will cleave the conjugates, followed by the efficient release of IMDQ, which activates the toll-like receptors 7/8 located in endo-lysosomes. By specific stimulation of TLR7/8 signalling, M2-like TAMs can be reprogrammed into M1-like phenotype with typical characteristics, including attenuated lysosomal acidity, decreased cathepsin activity, antigenic peptide preservation, and improved antigen cross-presentation for robust cancer immunotherapy. In contrast, $PGN_{4.9}$ nanoadjuvant keeps inert in other cell types within healthy tissues and circumvents acute systemic toxicity.

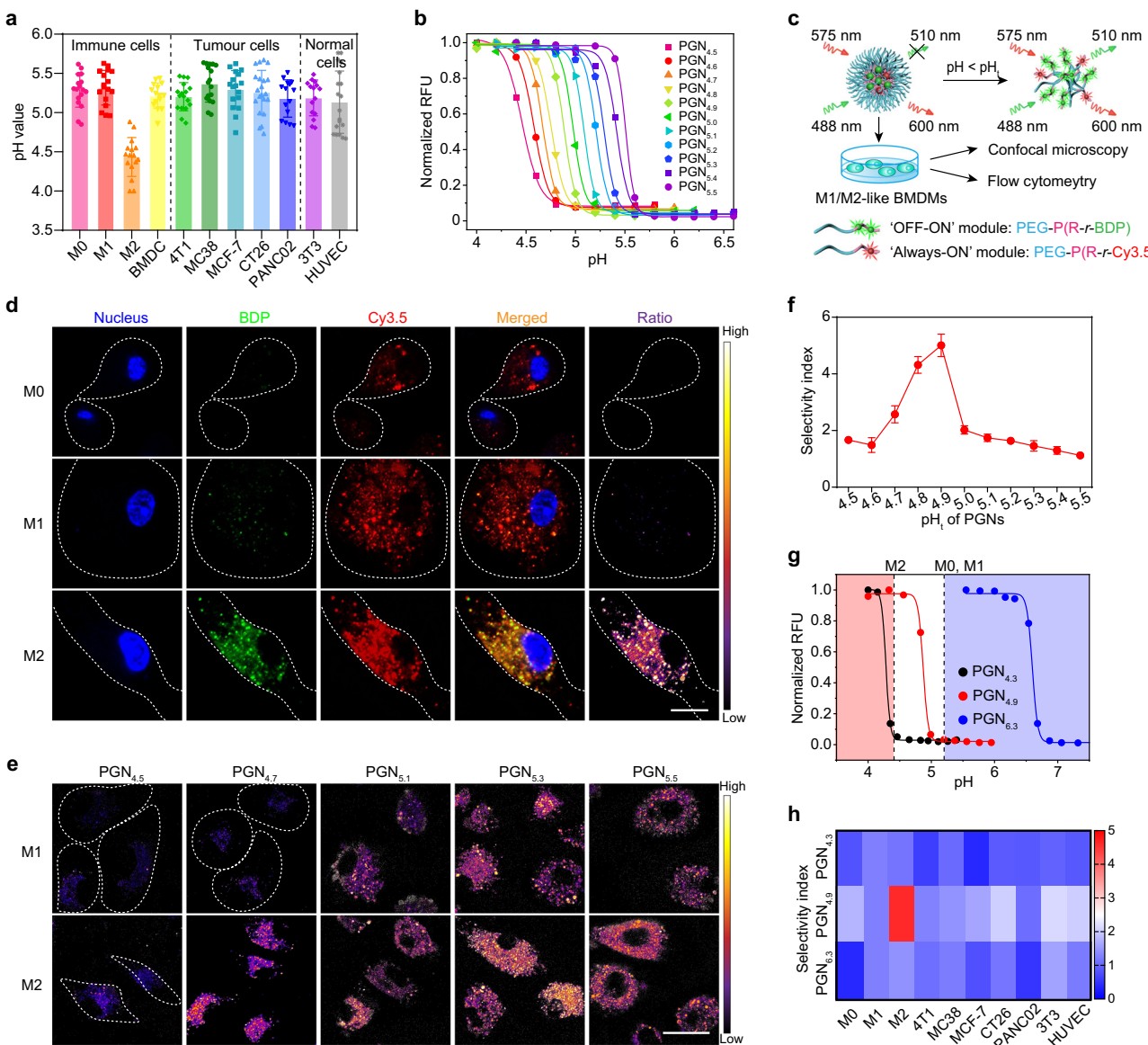

**Fig. 2 | Design of PGNs library that selectively responds to lysosomal pH of M2-like macrophages in vitro. a** The lysosomal pH value of various cell types (immune cells, tumour cells and normal cells) ($n = 15$ cells for BMDC, PANC02 and 3T3; $n = 16$ cells for M2; $n = 19$ cells for HUVEC; $n = 20$ cells for CT26; $n = 17$ cells for other groups). **b** PGNs library with a $pH_t$ ranging from 4.5 to 5.5 and pH increment of 0.1, covering the lysosomal pH range of different cell types ($n = 1$ experiment). **c** Schematic design of the PGNs library with 'Always-ON' and 'OFF-ON' fluorescence modules for ratiometric reporting of pH-gated dissociation. **d** Confocal images of macrophages with different phenotypes treated with PGN$_{4.9}$-BDP/Cy3.5 binary nanoreporter for 2 h. Ratiometric images represent the pH-gated activation of the nanoparticle independent of its concentration ($n = 3$ experiments). Green, BDP; red, Cy3.5; blue, nucleus; Scale bar, 10 μm. **e** Representative ratiometric images for the library of PGN-BDP/Cy3.5 binary nanoprobes with different $pH_t$ in a lysosomal environment of macrophages ($n = 3$ experiments). Scale bar, 20 μm. **f** The ratiometric readouts of M2-like macrophage normalised to that of M1-like macrophage as a function of $pH_t$ of PGNs ($n = 10$ cells). **g** PGN$_{4.9}$ nanoprobe can selectively be activated in lysosomal pH ($pH_L$ ~4.4) of M2-like macrophages, while keep silent in lysosomal pH ($pH_L$ ~5.2) of M0-, M1-like macrophages and other cells. In contrast, PGN$_{4.3}$ keeps 'OFF' and PGN$_{6.3}$ keeps 'ON' states in all cell types regardless of the interior acidity of lysosomes ($n = 1$ experiment). **h** Heat map showed the selectivity of three PGN nanoprobes to the lysosomal pH of different cell types ($n = 10$ cells). All measurements are presented as mean ± s.d. Source data are provided as a Source Data file.

which consisted of the 'OFF−ON' and 'Always-ON' fluorescence modules in visible-light window[24]. The 'Always-ON' modules keep fluorescent signal constant over pH changes that act as an internal standard to rule out the contribution of nanoparticle concentration to signal output, whereas the 'OFF−ON' modules exhibit more than 60-fold fluorescence amplification when pH drops below the pre-designed $pH_t$ for individual nanoparticle (Fig. 2c; Supplementary Fig. 5). After data processing, a ratiometric readout was obtained for binary reporting of the lysosomal pH of living cells. For nanoparticle characterisation, PGN$_{4.9}$ was chosen as an example that presented as

a spherical nanostructure with a diameter of 63.1 ± 2.6 nm at pH 7.4 while dissociated into unimer at pH 4.0 (Supplementary Fig. 6). The superior colloidal stability of PGN$_{4.9}$ in fresh mouse serum was confirmed by fluorimetry over a period of 24 hours (Supplementary Fig. 6e).

### PGN$_{4.9}$ renders pH-gated activation in the lysosome of M2-like macrophages in vitro

We first imaged the intracellular trafficking of PGNs after incubation with living cells for 2 h. Results showed that PGNs had a good

colocalization with lysosomes (Supplementary Fig. 7). To demonstrate the pH-gating capacity of PGNs, we investigated the signal activation inside lysosomal compartments of M0-, M1- and M2-like BMDMs by high-resolution fluorescence microscopy. Taking $PGN_{4.9}$ binary nanoreporter as an example, the BDP signal kept 'OFF' in the lysosomes of M0- and M1-like BMDMs, whereas it turned 'ON' in the lysosomes of M2-like ones. The 'Always-ON' Cy3.5 signal showed a perfect colocalization with the BDP signal in lysosomes of M2-like BMDMs (Fig. 2d). The ratiometric readouts of BDP to Cy3.5 signals kept 'OFF' in almost all lysosomes of M0- and M1-like BMDMs, whereas switched 'ON' in each lysosome of M2-like macrophages. The huge difference in ratiometric signals was succinctly defined as the selectivity index (SI), the ratio of ratiometric signals between the tested cells and M1-like BMDMs. We then systemically evaluated pH-gated activation of PGNs library with different $pH_t$ in M1- and M2-like BMDMs (Fig. 2e, Supplementary Fig. 8). The results demonstrated that $PGN_{4.9}$ exhibited up to 5-fold selectivity index towards the lysosomal pH of M2-like BMDMs relative to M1-like counterparts (Fig. 2f).

To further corroborate the selectivity of $PGN_{4.9}$ towards M2-like BMDMs, the pH-gated activation in several tumours and normal cell lines, including MC38 and CT26 colorectal cancer cells, MCF-7 breast cancer cells, PANC02 pancreatic cancer cell, NIH/3T3 mouse embryonic fibroblast, and human umbilical vein endothelial cell (HUVEC) were evaluated. The binary ratiometric nanoreporters of PGNs with $pH_t$ of 4.3 and 6.3, which are below and above the lysosome pH of all the cell lines, were chosen as control groups. As expected, $PGN_{4.3}$ indiscriminately kept 'OFF' and $PGN_{6.3}$ presented 'ON' in each lysosome of different cell types, respectively, whereas $PGN_{4.9}$ differentiated the lysosomal pH of M2-like macrophages (SI ~ 5) from those of M0- and M1-like macrophages, tumour cells and normal cells in vitro (Fig. 2g, 2h; Supplementary Figs. 9–11). The pH-gated activation of $PGN_{4.9}$ in M2-like BMDMs was also verified by flow cytometry (Supplementary Fig. 12).

### $PGN_{4.9}$ selectively profiles M2-like macrophages in vivo
To evaluate the lysosomal accumulation of PGNs in vivo, 4T1-tumour-bearing mice were intravenously injected with Cy5-conjugated $PGN_{4.9}$, and tumours were excised at 24 h post-injection and cryosectioned for the immunofluorescence staining of LAMP 1 (lysosome biomarker) in tumour slices. Results demonstrated that PGNs had a good colocalization with lysosomes within tumour tissues at 24 h post-administration (Supplementary Fig. 13). To evaluate the specific pH-gated activation in M2-like TAMs in vivo, PGNs were fluorescently labelled with indocyanine green (ICG) for near-infra-red imaging. $PGN_{6.3}$, $PGN_{4.9}$, and $PGN_{4.3}$ render more than 100-fold signal amplification when pH drops below the corresponding $pH_t$ of each nanoparticle (Supplementary Fig. 14). After intravenous injection, the fluorescent images of 4T1-tumour bearing mice were captured at predesignated time-points (Fig. 3a, Supplementary Fig. 15). Quantitative results showed that the fluorescent activation in $PGN_{4.9}$- and $PGN_{4.3}$-treated tumours was 2.23- and 0.87-fold of $PGN_{6.3}$-treated groups at 24 h post-administration, respectively (Fig. 3b). This high activation of $PGN_{4.9}$ in tumours was also evaluated in CT26-tumour bearing mice model (Supplementary Fig. 16). The tumour tissues were collected after in vivo imaging and homogenised to determine the absolute level of nanoparticle accumulation. As shown in Fig. 3c, the tumour accumulation in $PGN_{4.9}$- and $PGN_{4.3}$-treated groups was 1.93- and 2.06-fold of that in $PGN_{6.3}$-treated group, respectively. The ratiometric readouts of nanoparticle fluorescence activation versus absolute accumulation were calculated to report the efficiency of nanoparticle activation inside the lysosomal pH of cells within tumours. As expected, $PGN_{4.9}$ exhibited comparable tumour activation to $PGN_{6.3}$ but significantly higher efficient activation (~2.6-fold) as compared with $PGN_{4.3}$ (Fig. 3d). Intriguingly, the readouts of PGNs activation efficiency in mononuclear phagocyte system with abundant

naïve macrophages (e.g. liver and spleen) were significantly lower than that in tumour tissues (Fig. 3e, Supplementary Fig. 17). Collectively, the results demonstrated the pH-gated activation of $PGN_{4.9}$ in tumours rather than healthy normal tissues. Immunostaining of the whole-mount tumour sections revealed that more than 80% of $PGN_{4.9}$ selectively targeted M2-like TAMs rather than M1-like phenotype or tumour cells (Fig. 3f). In contrast, $PGN_{6.3}$ and $PGN_{4.3}$ manifested poor co-localisation with tumour cells or TAMs (Supplementary Fig. 18). Taken together, $PGN_{4.9}$ could selectively target intratumoural M2-like macrophages in vivo.

To further demonstrate the selectivity of $PGN_{4.9}$ towards M2-like macrophages in vivo, we established five tumour models with different macrophage contents, including CT26, MC38, 4T1, MCF-7 and PANC02. $PGN_{4.9}$ was covalently labelled with Cy5 and Cy7.5 as 'OFF–ON' and 'Always-ON' modules, respectively, for binary ratiometric imaging[24]. The ratiometric signals were collected at 24 h post-injection of $PGN_{4.9}$, and the proportion of $CD206^+F4/80^+$ M2-like macrophages in single cell suspension of tumour tissues was measured by flow cytometry (Supplementary Fig. 19). As shown in Fig. 3g, the curve of ratiometric signal (Cy5/Cy7.5) versus the percentage of M2-like macrophages exhibited an excellent linear correlation ($R = 0.7968$, $P < 0.001$). Moreover, depletion of 58.8% M2-like TAMs with clophosome[25] caused a significantly decreased signal activation of $PGN_{4.9}$, further elucidated the selective targeting of $PGN_{4.9}$ to M2-like TAMs (Fig. 3h–j; Supplementary Fig. 20). Besides, $PGN_{4.9}$ also had highest fluorescence signals in tumour-draining lymph nodes as compared with other groups, which were co-localised well with macrophages (Supplementary Fig. 21).

### $PGN_{4.9}$ nanoadjuvant re-educates M2-like macrophages to M1-like phenotype in vitro
Having elucidated the selectivity of $PGN_{4.9}$ towards M2-like TAMs, we proceeded to study its ability to precisely deliver immune agonists for the modulation of immunosuppressive macrophages. IMDQ, a TLR-7/8 agonist, was conjugated to the hydrophobic block of PGNs through Gly-Phe-Leu-Gly (GFLG) linkage. $PGN_{6.3}$ and pH-insensitive PEG-PEH copolymer conjugated with IMDQ through GFLG linkage were also synthesised (Supplementary Figs. 22 and 23; Supplementary Table 2). The polymer–drug conjugates can self-assemble into PGNs and non-pH-gated nanoadjuvant (NPGN). $PGN_{4.9}$ nanoadjuvant renders AND-gated behaviour that converts two orthogonal inputs (i.e. pH and cathepsin activity) into IMDQ release as a single output (Fig. 4a; Supplementary Fig. 24).

We then utilised a TLR reporter cell line to investigate the TLR-activating properties of IMDQ preparations in vitro. As shown in Fig. 4b, $PGN_{4.9}$ and $PGN_{6.3}$ nanoadjuvants exhibited significantly higher TLR activating activity than NPGN, as evaluated by the $EC_{50}$ assay and secretion of proinflammatory cytokine IL-12 (Supplementary Fig. 25a). The $EC_{50}$ of $PGN_{4.9}$ nanoadjuvant was one-order of magnitude lower than that of NPGN. Cell viability assay indicated IMDQ and $PGN_{6.3}$ nanoadjuvant exhibited higher cytotoxicity than other IMDQ preparations (Supplementary Fig. 25b–d). More importantly, $PGN_{4.9}$ nanoadjuvant greatly decreased the undesired cytokine storm evoked by intravenous injection of free IMDQ, as verified by the diminished acute systemic abundant secretion of IP-10 and IL-12 in serum at 3 h post-injection. Hematoxylin and eosin (H&E) staining of major organs showed that pretreated mice kept tissue structure intact without cell morphology change in major organs, indicating good biocompatibility of $PGN_{4.9}$ nanoadjuvant (Supplementary Fig. 26). However, $PGN_{6.3}$ nanoadjuvant caused excessive activation of the systemic immune system probably due to the non-specific uptake and undesired release of IMDQ in immunocytes from bloodstream and normal tissues (Supplementary Fig. 25e–h). In consideration of animal ethics, $PGN_{6.3}$ nanoadjuvant was excluded from further experiments in vitro and in vivo.

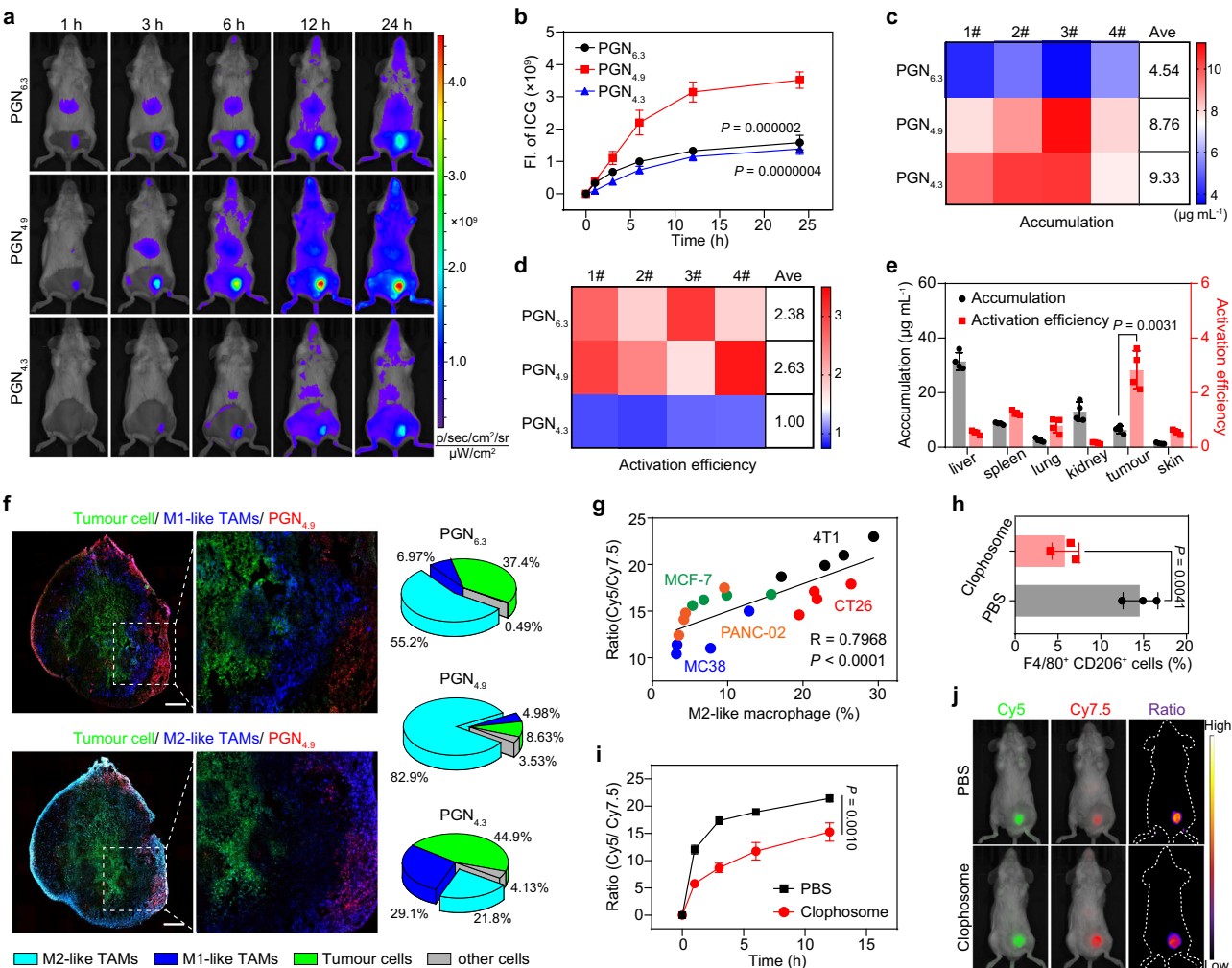

**Fig. 3 | PGN_{4.9} selectively targets to M2-like TAMs in vivo. a** Representative fluorescence images and **b** fluorescence intensity of tumour sites at different time-points after intravenous injection of ICG-conjugated PGNs (20 mg kg$^{-1}$) in 4T1 tumour-bearing mice ($n = 4$ mice; two-way ANOVA followed by Tukey's multiple comparisons test). **c** Heat map showed the tumour accumulation and **d** relative activation efficiency of three ICG-conjugated PGNs in 4T1 tumour model ($n = 4$ mice). **e** Accumulation level and activation efficiency of PGN_{4.9} in dissected livers, spleens, lungs, kidneys, tumours and skin ($n = 4$ mice, one-way ANOVA followed by Tukey's multiple comparisons test). **f** Whole-mount images of tumour adjacent slices from 4T1-GFP tumour-bearing mice at 24 h post-administration of Cy5-labelled PGNs. Scale bar, 500 μm. The M1-like and M2-like macrophages were stained with anti-CD86 and anti-CD206 antibodies, respectively. The proportions of tumour cells, M1- and M2-like macrophages among the PGN-positive cells within tumour tissues ($n = 3$ experiments). **g** Linear correlation between ratiometric signals versus percentage of M2-like macrophages in different tumour models (4T1, MCF-7, CT26, MC38 and PANC02) treated with PGN_{4.9}-Cy5/Cy7.5 binary nanoreporter ($n = 4$ mice per group). **h** Depletion and repopulation of TAMs in 4T1-bearing mice after intravenous injection of clophosome for three times ($n = 3$ mice; two-tailed unpaired Student's $t$-test). **i** Time-dependent ratiometric signals of Cy5 to Cy7.5 in PBS- and clophosome-treated tumour tissues ($n = 4$ mice for PBS group; $n = 3$ mice for clophosome group; two-way ANOVA followed by Tukey's multiple comparisons test). **j** Ratiometric images of tumour-bearing mice at 6 h post-administration of PGN_{4.9} in PBS- and clophosome-treated animals. All measurements are presented as mean ± s.d. Source data are provided as a Source Data file.

Cell morphology has been considered as a unique biomarker of cell function[26]. The shape of macrophages was monitored before and after treatment with PGN_{4.9} nanoadjuvant. Confocal images showed that IL-4-treated BMDMs present a spindle shape, a characteristic of M2-like macrophages. After treatment with PGN_{4.9} nanoadjuvant for 24 h, the cell morphology changed into fried egg type, a characteristic of M1-like macrophages (Supplementary Fig. 27). Shotgun proteomics analysis identified 25 proteins were significantly changed in PGN_{4.9} nanoadjuvant pretreated M2-like BMDMs, consistent with the cell lysates in M1-like phenotype (Fig. 4c). Compared with PBS and NPGN, PGN_{4.9} nanoadjuvant significantly enhanced the typical protein expression of M1-related CD86 and iNOS, while decreased the expression of M2-related CD206 and arginase (Fig. 4d, e; Supplementary Fig. 28). Thus, PGN_{4.9} nanoadjuvant could be employed as a powerful driver of macrophage repolarization in vitro. Moreover, PGN_{4.9} nanoadjuvant distinctly decreased the ratiometric signals of

PGN_{4.9}-BDP/Cy3.5 binary nanoreporter in M2-like macrophages, indicating an attenuated lysosomal acidity upon TLR activation (Fig. 4f; Supplementary Fig. 29).

## PGN_{4.9} nanoadjuvant promotes antigen processing and presentation via regulation of lysosomal function

Having validated the reprogramming of macrophage phenotypes by PGN_{4.9} nanoadjuvant in vitro, we further explored antigen cross-presentation of BMDMs upon PGN_{4.9} nanoadjuvant treatment (Fig. 4g). Strikingly, PGN_{4.9} nanoadjuvant pretreatment alkalise the lysosomal steady-state pH from 4.52 to 5.22 in M2-like BMDMs, which is comparable to that in M1-like phenotype (pH_L ~5.23) and IMDQ-treated positive control (pH_L ~5.10), as determined by the ratiometric pH quantification. In contrast, NPGN nanoadjuvant failed to induce pH fluctuations (pH_L ~4.53) (Fig. 4h; Supplementary Fig. 30). It has been reported that lysosomal protease activity is closely related to

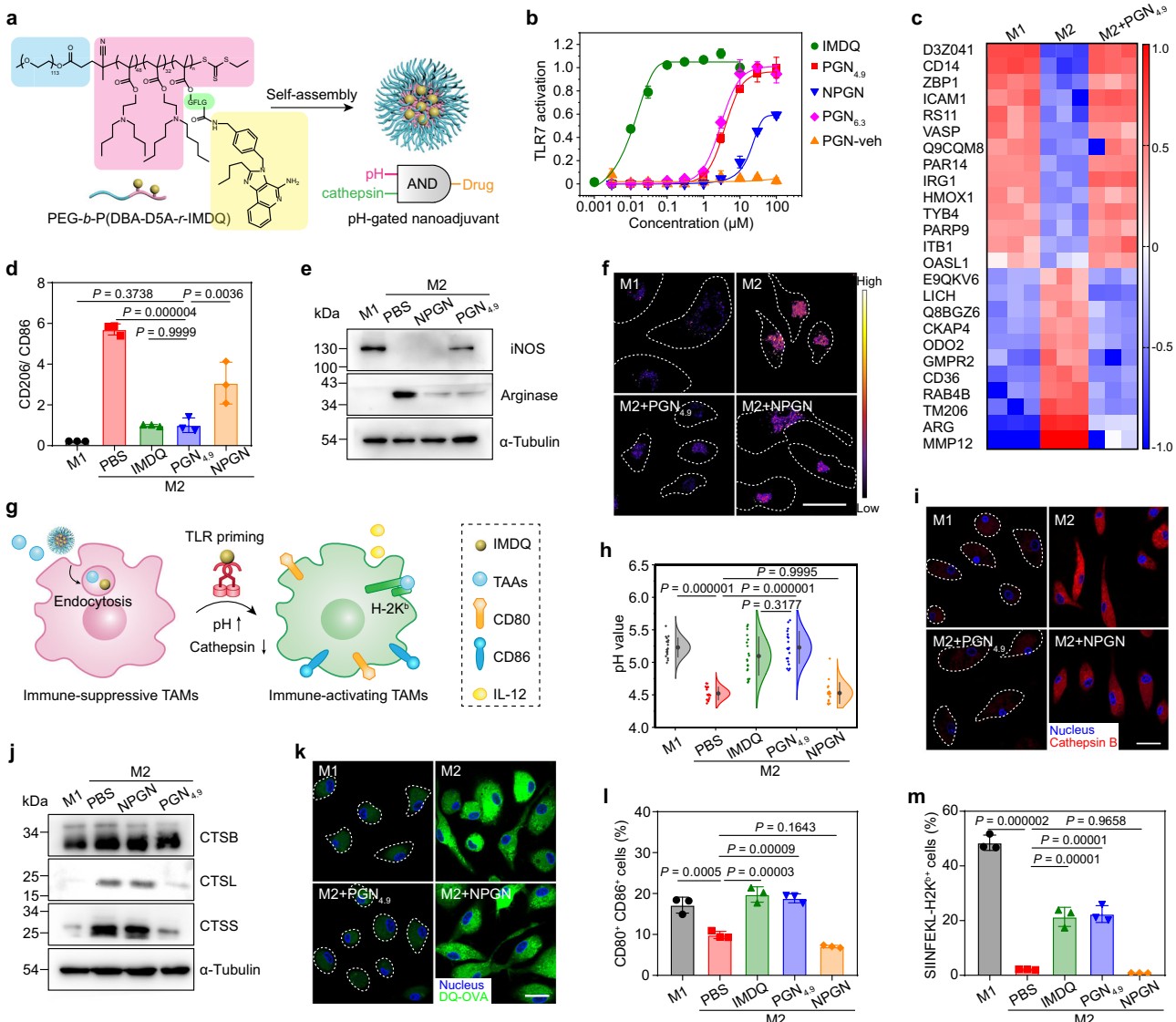

**Fig. 4 | PGN$_{4.9}$ nanoadjuvant reprogrammes M2-like macrophages into M1 phenotype for enhanced antigen cross-presentation in vitro. a** AND-gated design of PGN$_{4.9}$ nanoadjuvant with pH- and enzyme-dual selectivity. **b** TLR agonistic activity of various IMDQ treatments via a TLR reporter cell assay ($n=6$ experiments). **c** Heatmap of several protein levels from proteomics data of in M1-, M2- and M2-like BMDMs treated with PGN$_{4.9}$ nanoadjuvant ($n=3$ experiments). Scale, (M1 − M2$_{avg}$)/(M1 + M2$_{avg}$), (M2 − M1$_{avg}$)/(M2 + M1$_{avg}$) or (PGN$_{4.9}$ − M2$_{avg}$)/(PGN$_{4.9}$ + M2$_{avg}$). **d** Ratio of CD206 to CD86 expression in BMDMs lysate by flow cytometry ($n=3$ experiments). **e** The expression of iNOS and Arginase in M1-like BMDMs or M2-like phenotype treated with IMDQ preparations for 24 h ($n=3$ experiments). **f** Ratiometric images of M2-like macrophages treated with different IMDQ formulations, followed by the treatment of PGN$_{4.9}$ nanoreporter ($n=3$ experiments). Scale bar, 20 μm. **g** Schematic illustration of PGN$_{4.9}$ nanoadjuvant resetting M2-like BMDMs to M1 phenotype for antigen cross-presentation.

**h** The lysosomal pH of M2-like macrophages after incubation with various IMDQ formulations ($n=18$ cells for M2 group; $n=19$ cells for M1, IMDQ and PGN$_{4.9}$ groups; $n=20$ cells for other groups). **i** Reduced expression of cathepsin B in M2-like macrophage after treatment with PGN$_{4.9}$ nanoadjuvant ($n=3$ experiments). Scale bar, 20 μm. **j** Immunoblots of representative cathepsin family (e.g. cathepsin B, L and S) in M2-like BMDMs before and after treatment with PGN$_{4.9}$ nanoadjuvant ($n=3$ experiments). **k** Representative images of DQ-ovalbumin (green) degradation assays in M2-like BMDMs pretreated with PGN$_{4.9}$ nanoadjuvant. Scale bar, 20 μm. **l** CD80$^+$CD86$^+$- and **m** antigen-positive BMDMs after treatment with different IMDQ formulations and OVA$_{257-280}$ long peptides for 24 h ($n=3$ experiments). Statistical significance was analysed by one-way ANOVA followed by Tukey's multiple comparisons test. All measurements are presented as mean ± s.d. Source data are provided as a Source Data file.

enhanced lysosomal acidification of different cell types[27] and lysosomal proteolysis in antigen-presenting cells negatively correlates with antigen cross-presentation[12,28]. Using Cathepsin B activity assay and western blot, we found that BMDMs pretreated with PGN$_{4.9}$ nanoadjuvant exhibited remarkably attenuated lysosomal cathepsin activity and protease contents as compared with antigen-destroying M2-like BMDMs (Fig. 4i, j). These results highlight the crucial roles of PGN$_{4.9}$ nanoadjuvant in reprogramming M2-like TAMs to M1-like phenotype by regulation of lysosomal function (i.e. pH and proteolysis capacity).

Next, we evaluated the lysosomal degradation capacity using an ovalbumin degradation assay (DQ-OVA)[29]. As shown in Fig. 4k, PGN$_{4.9}$ nanoadjuvant-treated BMDMs attenuated the degradation ability of lysosomal proteases. Conversely, M2-like BMDMs displayed stronger lysosomal proteolysis, which easily led to antigen destruction (Fig. 4k and Supplementary Fig. 31). Upregulation of costimulators, antigen presentation and secretion of proinflammatory cytokines contribute to cytotoxic T cell activation[30]. The proportion of CD80$^+$CD86$^+$ BMDMs was significantly enhanced after M2-like BMDMs pretreated

with $PGN_{4.9}$ nanoadjuvant (Fig. 4l and Supplementary Fig. 32). Moreover, $PGN_{4.9}$ nanoadjuvant caused a significantly higher MHC Class I molecules on the cell surface of M2-like BMDMs after repolarization to M1-like phenotype (Supplementary Fig. 33). Meanwhile, $PGN_{4.9}$ nanoadjuvant pretreatment allowed efficient $OVA_{257-280}$ and $OVA_{257-264}$ presentation with strong MHC class I-associated SIINFEKL displayed on the cell membrane of BMDMs (Fig. 4m and Supplementary Fig. 34), although the extended peptides need intracellular processing to suitable size for cross-presentation[31]. Moreover, co-inoculation of 4T1 tumour cells and re-educated M2-like BMDMs in BALB/c mice showed that $PGN_{4.9}$ nanoadjuvant significantly retarded tumour formation and prolonged the animal survival (Supplementary Fig. 35). In summary, $PGN_{4.9}$ nanoadjuvant could efficiently reprogramme M2-like macrophages via regulation of lysosomal function for enhanced antigen cross-presentation and tumouricidal capacity.

## $PGN_{4.9}$ nanoadjuvant promotes an ameliorative tumour microenvironment

To profile tumour microenvironment modulation by $PGN_{4.9}$ nanoadjuvant, tumour-infiltrated lymphocytes were visualised by multi-colour immunohistochemistry staining (Fig. 5a, b). Unlike control groups, $PGN_{4.9}$ nanoadjuvant-treated mice presented higher intratumoural ratios of M1-like TAMs to M2 subset, implying the reprogramming capability of $PGN_{4.9}$ nanoadjuvant in vivo (Fig. 5c; Supplementary Fig. 36). Consistently, $CD8^+$ T cells exhibited an elevated proliferation potential (Fig. 5d) and the proportion of $CD4^+Foxp3^+$ regulatory T cells were downregulated after $PGN_{4.9}$ nanoadjuvant treatment (Supplementary Fig. 37). $PGN_{4.9}$ nanoadjuvant significantly promoted the immune-supportive components, while decreased the immune-suppressive populations in tumour microenvironment.

We investigated the antigen cross-presentation by macrophages in MC38.OVA tumour-bearing mice upon intravenous injection of $PGN_{4.9}$ nanoadjuvant. The pre-immunised mice were euthanized and the tumour-draining lymph nodes and tumour tissues were harvested for the following experiment. The expression of costimulatory markers (such as CD80 and CD86) on lymph node-resident macrophages was remarkably upregulated in $PGN_{4.9}$ nanoadjuvant-treated mice as compared with control mice (Fig. 5e; Supplementary Fig. 38a–c). Meanwhile, $PGN_{4.9}$ nanoadjuvant also induced an increased presentation of cell-surface SIINFEKL peptide on macrophages from tumour-draining lymph nodes and intratumoural TAMs of immunised mice (Fig. 5f; Supplementary Fig. 38d–f). Thus, $PGN_{4.9}$ nanoadjuvant promoted effective antigen cross-presentation in vivo.

To further corroborate the activation of cytotoxic $CD8^+$ T cells upon the immunomodulation of $PGN_{4.9}$ nanoadjuvant in vivo, we evaluated the antigen-specific CTL response by specific splenocyte killing experiment in immunised MC38.OVA-bearing mice. As shown in Fig. 5g, h, $PGN_{4.9}$ nanoadjuvant achieved the most efficient specific killing (92.4%) as compared with other groups. Consistently, the intravenous injection of $PGN_{4.9}$ nanoadjuvant increased 4.7-fold IFN-γ production from CTLs with the capacity to kill tumour cells (Fig. 5i, j). Therefore, $PGN_{4.9}$ nanoadjuvant induced a robust antigen-specific immune response to stimulate $CD8^+$ T cell activation and proliferation efficiently. Moreover, we observed increases in effector memory T cells (Tem) and central memory T cells (Tcm) in $CD8^+$ and $CD4^+$ T cell subsets in splenocytes from MC38 tumour-bearing mice treated with $PGN_{4.9}$ nanoadjuvant (Fig. 5k–m; Supplementary Fig. 39). Thus, the $PGN_{4.9}$ nanoadjuvant immunotherapy not only brought an ameliorative tumour microenvironment but also provided a specific long-acting immune protection.

## $PGN_{4.9}$ nanoadjuvant selectively targets M2-like TAMs to promote anti-tumour immunity

Next, we assessed the anti-tumour efficacy of $PGN_{4.9}$ nanoadjuvant (Fig. 6a). In 4T1-luciferase breast cancer model, $PGN_{4.9}$ nanoadjuvant

treatment caused a remarkable tumour regression with an inhibition rate of 78.6%, which was significantly higher than other groups (Fig. 6b–e; Supplementary Fig. 40). In accordance with the therapeutic activity, $PGN_{4.9}$ nanoadjuvant also circumvented the lung metastasis of 4T1 tumours noticeably (Fig. 6f, g; Supplementary Fig. 41).

To certify the significance of TAMs in $PGN_{4.9}$ nanoadjuvant mediated tumour attenuation, CSF1R antibody was injected intraperitoneally to deplete macrophages in orthotopic 4T1 tumour-bearing mice[32] (Supplementary Fig. 42). The effects of $PGN_{4.9}$ nanoadjuvant on tumour inhibition was significantly mitigated by depletion of 76.8% macrophages in tumour-bearing mice (Fig. 6h). T cell depletion also led to tumour relapse in mice treated with $PGN_{4.9}$ nanoadjuvant (Supplementary Fig. 43). Since $PGN_{4.9}$ nanoadjuvant enabled TAMs to efficiently present tumour-associated antigen for tumour regression, we next asked whether enhanced antigen production by triggering cancer cell death could further improve the anti-tumour immunity. Combining with PDPA-DTX nanomedicine developed by our laboratory[33], $PGN_{4.9}$ nanoadjuvant exhibited a greater tumour inhibition and prolonged survival with a reduced adjuvant dosage through boosting intratumoural cytotoxic T cells, as compared with nanoadjuvant treatment alone (Fig. 6i, j; Supplementary Fig. 44). To further examine the applicability of $PGN_{4.9}$ nanoadjuvant, MC38 colorectal tumour-bearing mice were immunised with different preparations, and $PGN_{4.9}$ nanoadjuvant immunotherapy achieved prominent anti-tumour efficacy (Supplementary Fig. 45). After the anti-tumour immunotherapy, tumour-free mice in $PGN_{4.9}$ nanoadjuvant group received first and second MC38 tumour rechallenge on day 73 and day 120, respectively. The immunised mice successfully resisted the multiple tumour attacks, demonstrating that $PGN_{4.9}$ nanoadjuvant provided long-term protection beyond 150 days (Fig. 6k).

Considering that effective re-education of macrophages from M2-like phenotypes to M1-like provoked adaptive immunity through cytotoxic T cell activation, we inferred that $PGN_{4.9}$ nanoadjuvant could potentiate the therapeutic efficacy of immune checkpoint blockade. We combined $PGN_{4.9}$ nanoadjuvant with α-PD1 for the treatment of MC38 tumour-bearing mice and monitored the tumour growth over 70 days. The results showed that α-PD1 treatment alone led to no significant inhibition of tumour growth, whereas 70% of the mice immunised with α-PD1 and $PGN_{4.9}$ nanoadjuvant exhibited no tumour progression within 70 days by significantly promoting the intratumoural infiltration of $CD8^+$ T cells (Fig. 6l–n; Supplementary Fig. 46). Taken together, combination therapy could offer a possibility to achieve synergistic innate and adaptive immunity for robust therapeutic efficacy.

## Discussion

TAMs represent a heterogeneous population with distinct functions across many cancer types[3]. It is necessary to understand the dynamic interactions between TAMs and other infiltrated immune cells at the different stages of tumour progression[34]. Currently, some anti-TAMs drugs are under preclinical and clinical evaluation, consisting of three main strategies[4]: (i) inhibition of TAMs recruitment[35], (ii) TAMs depletion[36], and (iii) re-education of M2-like TAMs[37]. However, previous studies have demonstrated that the inhibition of TAMs recruitment and survival might not suffice to stimulate durable anti-tumour response[10], whereas the re-education strategy represents a more effective choice to not only ameliorate the immunosuppressive functions but also to potentiate antigen cross-presentation.

Adaptive immune response requires effective antigen cross-presentation, which is regulated by the appropriate lysosomal activity of professional antigen-presenting cells to maximise the production of a suitable length of antigen peptides, whereas hyperactive lysosomal proteolysis easily causes the high degradation of antigen[12]. As for TAMs, their lysosomal degradative capacity relies on the lysosomal acidity of opposing phenotypes[8,27], wherein M1-like macrophages have

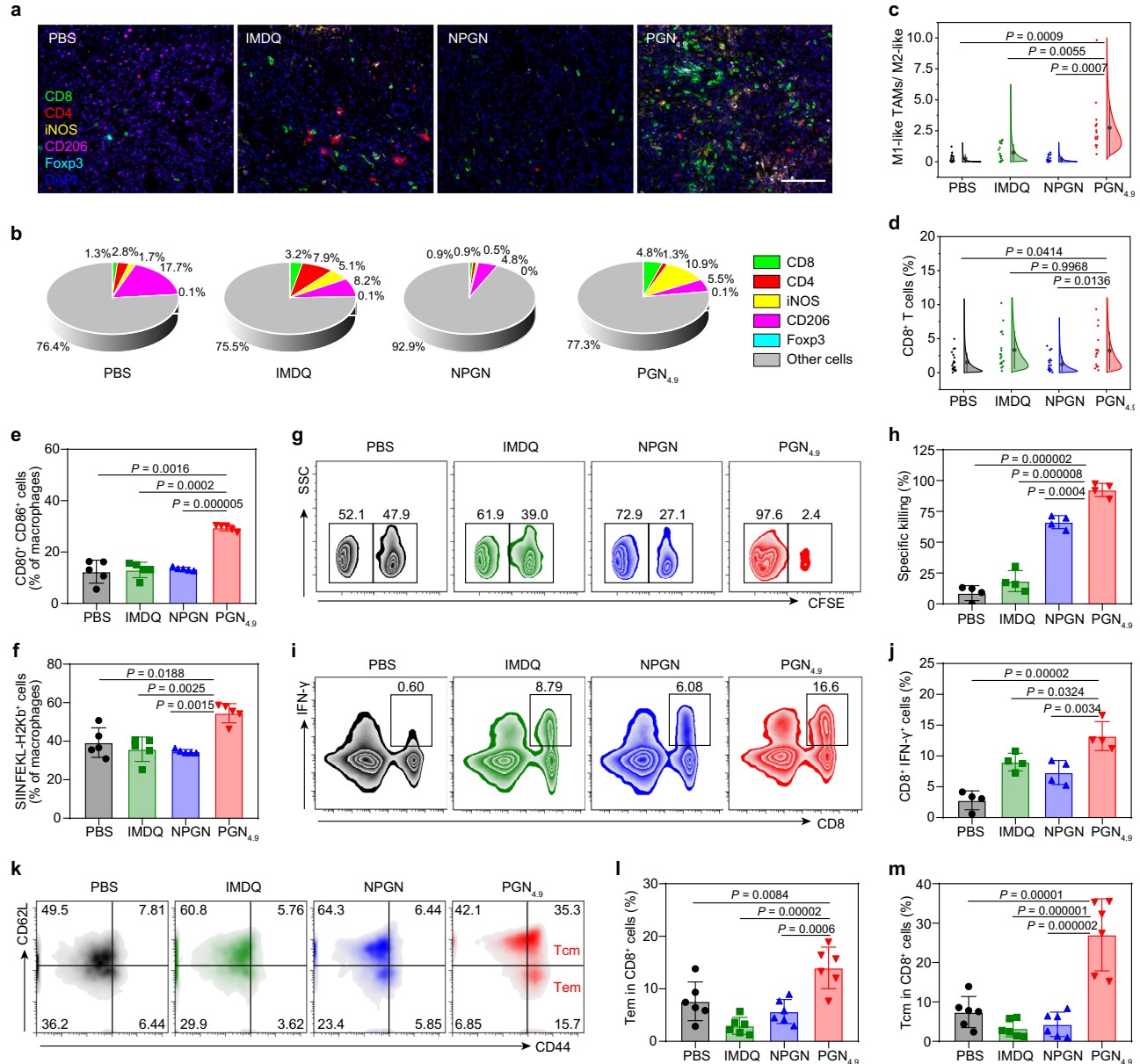

**Fig. 5 | PGN_{4.9} nanoadjuvant ameliorates immunosuppressive TME. a** Multi-colour immunohistochemistry staining of tumour tissues from 4T1 tumour-bearing mice after treatment with PBS and different IMDQ formulations, emphasising CD8, CD4, iNOS (M1-like macrophage marker), CD206 (M2-like macrophage marker) and Foxp3 (n = 3 experiments). Scale bar, 100 μm. **b** Corresponding pie chart showing the proportions of various lymphocytes in the tumour microenvironment (n = 3 experiments). **c** The ratio of M1-like macrophages to M2 phenotype and **d** CD8+ T cells from images in panel (**a**) (n = 16 regions for PGN_{4.9} group; n = 18 regions for IMDQ group; n = 20 regions for other groups). **e** CD80+CD86+ macrophages in tumour-draining inguinal and popliteal lymph nodes from MC38.OVA tumour-bearing mice upon intravenous injection of IMDQ formulations (n = 5 mice). **f** Effect of nanoadjuvants on antigen cross-presentation by macrophages from MC38.OVA tumour-bearing mice were intravenously treated with IMDQ, NPGN and PGN_{4.9} nanoadjuvants (n = 5 mice). **g** Representative flow cytometry plots and **h** quantification of OVA-specific CTL killing capacity in MC38.OVA-bearing mice (n = 4 mice). **i, j** The percentage of CD8+IFN+ splenocytes from mice with different treatments (n = 4 mice). **k** Representative scatterplots of CD44^{high} CD62L^{low} T cell and CD44^{high} CD62L^{high} T cell subsets among CD8+ T lymphocytes. **l** Effector memory T cells and **m** central memory T cells of CD8+ T cells from MC38 tumour-bearing mice in different treatments (n = 6 mice). Statistical significance was analysed by one-way ANOVA followed by Tukey's multiple comparisons test. All measurements are presented as mean ± s.d. Source data are provided as a Source Data file.

an optimal lysosomal activity that triggers the subsequent antigen cross-presentation for the activation of CD8+ T cell function. Thus, specific polarization of M2-like TAMs to M1-like phenotype holds great promise to reverse immunosuppressive tumour microenvironment, enhance antigen cross-presentation, and finally achieve robust cancer immunity.

Immunoagonists (e.g. R848 and CpG) in the free form are easily distributed throughout the body and cause severe systemic immunotoxicity, which hampers their clinical translation[38,39]. A variety of nanoparticles have been developed to deliver immunoagonists and achieve the polarization of M2-like TAMs[10,40,41]. Recent studies demonstrated that intravenous delivery might increase the possibility of effective co-localisation of immunologic adjuvant with dying tumour cells, thus producing an in situ vaccination for superior immune responses[42]. However, many such strategies are based on the tissue targeting mechanism that could also activate M0- and M1-like macrophages in non-malignant organs, including the liver, spleen, lung and skin, raising biosafety concerns. Therefore,

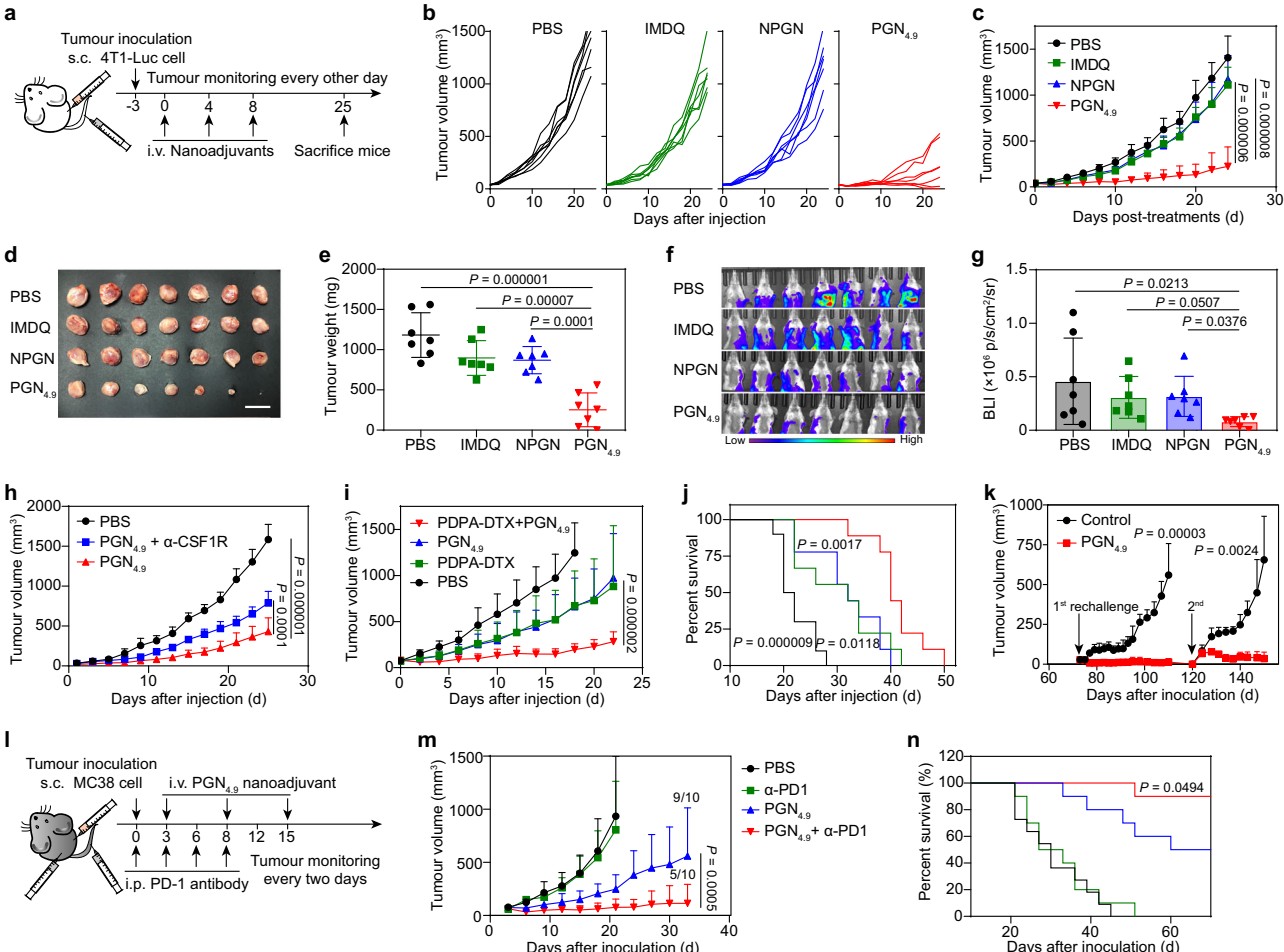

**Fig. 6 | In vivo therapeutic efficacy of PGN$_{4.9}$ in different tumour models.**
**a** Schematic illustration for 4T1-luciferase tumour immunotherapy. **b** Individual tumour growth kinetics and **c** average tumour growth curves of 4T1-luc tumour-bearing mice treated with PBS, free IMDQ, NPGN and PGN$_{4.9}$ (equivalent to 2 mg kg$^{-1}$ IMDQ, $n = 7$ mice). **d** Photographs and **e** weight of excised tumours ($n = 7$ mice, one-way ANOVA followed by Tukey's multiple comparisons test). Scale bar, 2 cm. **f** Bioluminescence imaging and **g** corresponding quantification of lung metastases after intravenous administration ($n = 7$ mice, one-way ANOVA followed by Tukey's multiple comparisons test). **h** Tumour immunotherapy after macrophage depletion in 4T1 tumour-bearing mice ($n = 5$ mice for combination group; $n = 6$ mice for other groups). **i** On day 7, BALB/c mice were inoculated subcutaneously with 4T1 cells. Tumour progression and **j** survival curves of 4T1 tumour-bearing mice treated with PGN$_{4.9}$ nanoadjuvant (equivalent to 1 mg kg$^{-1}$

IMDQ) and PDPA-DTX nanomedicine (equivalent to 3 mg kg$^{-1}$ DTX), alone or in combination on day 0, 4, 8 and 12 ($n = 9$ mice). **k** On day 73 and day 120, PGN$_{4.9}$ nanoadjuvant-treated mice in the MC38 colorectal tumour model were rechallenged with MC38 cells ($2 \times 10^6$ cells/mouse). For the control groups, naïve C57BL/6 mice were subcutaneously injected with the same number of MC38 cells ($n = 9$ mice at day 73 and $n = 6$ mice at day 120). **l** Schematic schedule of combined immunotherapy of α-PD1 and PGN$_{4.9}$. **m** Tumour progression and **n** survival curves of MC38 tumour-bearing mice treated with PBS, α-PD1 (100 μg per dose), PGN$_{4.9}$ nanoadjuvant (equivalent to 2 mg kg$^{-1}$ IMDQ) and combined administration ($n = 11$ mice for PBS group; $n = 10$ mice for other groups). The long-term survival was calculated by a log-rank test. Other statistical significance was analysed by two-way ANOVA followed by Tukey's multiple comparisons test. All measurements are presented as mean ± s.d. Source data are provided as a Source Data file.

nanoadjuvant technology that specifically modulates M2-like TAMs would provide safe and effective formulations optimised for cancer immunotherapy.

Based on our finding that the lysosomal pH difference between M2-like TAMs (pH$_L$ ~4.4) and M0- and M1-like macrophages (pH$_L$ ~5.2), engineering of pH-responsive nanotechnology would be a promising approach to achieve targeted modulation of M2-like TAMs. So far, pH-responsive nanoparticles have been extensively reported to shuttle the therapeutic cargoes to solid tumours through pH-triggered nanocarrier disintegration or linkage cleavage upon pH changes within the tumour microenvironment[43–45]. Although these approaches offer the targeted delivery of therapeutic cargoes to lysosomal compartments, achieving selective targeting to the highly lysosomal acidity of M2-like TAMs rather than that of other cells remains a significant challenge. In this article, our PGN nanoadjuvants were successfully designed and screened with several key features that enable the specific

polarization of M2-like TAMs instead of other macrophages in normal tissues due to their AND-gated performance. Firstly, the PGN nanoadjuvants render a sharp pH response (ΔpH$_{ON/OFF}$ ~ 0.2–0.3), which is critical for the pH-gated activation in the lysosomal compartment of M2-like TAMs rather than the counterpart of other macrophages. Secondly, the pH$_t$ tunability enables the successful screening of PGN$_{4.9}$ nanoadjuvant for the specific targeting of lysosomal pH of M2-like TAMs, followed by enzymatic cleavage-mediated drug release to achieve logic-gated immunotherapies.

In summary, we have successfully developed a pH-gated nanoadjuvant with AND-gated capacity that can selectively fine-tune lysosomal proteolysis of M2-like TAMs to facilitate antigen cross-presentation and provoke adaptive tumour immunity. Our strategy provides a powerful toolbox for specific targeting and stimulating distinct signalling pathways within specific endocytic organelles to advance cancer immunotherapy.

## Methods

### Ethical statement
This research complies with all relevant ethical regulations. All animal studies were conducted in accordance with the National Institute Guide for the Care and Use of Laboratory Animals. The experimental protocols were approved by the Institutional Animal Care and Use Committee (IACUC) of Peking University (Accreditation number: LA 2019039).

### Materials
The BDP-NHS, Cy3.5-NHS, Cy5-NHS and Cy7.5-NHS esters were purchased from Lumiprobe Company. ICG-Sulfo-OSu was obtained from AAT Bioquest Inc. Monomers 2-(diisopropylamino) ethyl methacrylate (iDPA-MA) and 2-aminoethyl methacrylate (AMA) were purchased from Polyscience Company. Monomers 2-(dipropylamino) ethyl methacrylate (nDPA-MA), 2-(dibutylamino) ethyl methacrylate (DBA-MA), 2-(dipentylamino) ethyl methacrylate (D5A-MA), $PEG_{5k}$-CTA, MA-GFLG-IMDQ were synthesised in our laboratory[22,46]. Imidazoquinoline (IMDQ) was obtained from Nanjing Aikon Chemical Ltd.

### Cell lines
4T1 (3101MOUSCSP5056) and MCF-7 breast cancer cells (3101HUMSCSP531), CT26 (1101MOU-PUMC000275), PANC02 pancreatic cancer cell (CRL-2553), RAW264.7 macrophage (3101MOUSCSP5036), NIH/3T3 mouse embryonic fibroblast (1101MOU-PUMC000018), and human umbilical vein endothelial cell (HUVEC, 4201PAT-CCTCC00692) were obtained from National Infrastructure of Cell Line Resource. TLR reporter cell line, RAW-Blue (raw-sp), was purchased from InvivoGen. MC38 colorectal cancer cells and MC38.OVA cells were obtained from Dr. Wei Liang (Institute of Biophysics, Chinese Academy of Sciences). The 4T1-GFP cell line was provided by Dr Yucai Wang (School of Life Sciences, University of Science and Technology of China). These cell lines were cultured in RPMI 1640 (for 4T1, 4T1-GFP, MCF-7 and CT26), DMEM (for PANC02 and MC38) or F12k (for HUVEC) supplemented with 10% FBS and antibiotics under 5% CO2 atmosphere at 37 °C. RAW264.7 and RAW-Blue cells were cultured with a heat-inactivated DMEM medium. In addition, murine bone marrow-derived macrophages (BMDMs) were obtained by isolating bone marrow cells from surgically resected femur and tibia of C57BL/6 female mice (6–8 weeks). M0-like BMDMs were cultured in the complete heat-inactivated DMEM medium containing recombinant murine macrophage CSF (m-CSF, 20 ng mL$^{-1}$, PeproTech) for 5 days. M0-like BMDMs were stimulated with lipopolysaccharide (LPS, 100 ng mL$^{-1}$) and interferon γ (IFN-γ, 20 ng mL$^{-1}$, PeproTech) or IL-4 (20 ng mL$^{-1}$, BIOplastics) for 24 h to obtain M1-like BMDMs or M2-like phenotypes, respectively.

### Animal models
Female BALB/c (6–8 weeks), C57BL/6 (6–8 weeks), and nu/nu nude mice (6–8 weeks) were sourced from Vital River Laboratory Animal Centre (Beijing, China) and maintained in specific pathogen-free (SPF) conditions for 1 week before the studies. The animals were housed at a temperature of 25 °C in a humidity-controlled environment with free access to food and water in a 12 h light/dark cycle. To establish an orthotopic 4T1 tumour model, as an example, 4T1 cells ($1 \times 10^6$ cells/mouse) were inoculated subcutaneously on the right mammary fat pad of BALB/c mice. When tumour volume reached about 100 mm³, the tumour-bearing mice were used for the following in vivo fluorescence imaging study and anti-tumour study. For anti-tumour immunotherapy, the maximal tumour size of 1500 mm³ was permitted by IACUC of Peking University. In some cases, this limit has been exceeded by the last day of measurement, and the mice were immediately euthanized.

### Measurement of lysosomal pH
To determine the lysosomal pH of different cell lines, an in situ pH calibration curve was established following the reported procedure[19].

Firstly, Oregon Green-dextran (Invitrogen) and Rhodamine B-dextran (Sigma) were dissolved in DMEM medium at 5 mg mL$^{-1}$ at a molar ratio of 1:1. Cell lines were pulsed with mixed fluorescent dextran for 6 h, chased for 12 h in fresh medium, and washed with PBS to calculate the fluorescence intensity of lysosomes by a confocal microscope (A1R-Storm, Nikon) under 60× oil objective lens. Subsequently, a corresponding calibration was performed for each lysosome. The cells were incubated with nigericin (10 μM) in high-K$^+$ buffers from pH 4.0 to 6.0 for 5 min equilibrium on the ice. The fluorescence intensity was measured by the confocal microscope. The Hoechst 33342 (Invitrogen), Oregon Green, and Rhodamine B were excited at 405, 488 and 561 nm, respectively. The ratiometric images were processed by NIS-Elements viewer software, and the resulting quantification was calculated by ImageJ software (NIH), which was plotted as a function of pH value and fitted to a Boltzmann sigmoid to measure the lysosomal pH in various cell lines.

### Syntheses and characterisation of PEG-$b$-(PR-$r$-AMA$_3$) copolymers
PEG-$b$-(PR-$r$-AMA$_3$) copolymers were synthesised via the atom transfer radical polymerisation (ATRP) method. Taking PGN$_{4.9}$ as an example, $PEG_{5k}$-Br (0.5 g, 0.1 mmol), DBA-MA (1.16 g, 4.8 mmol), D5A-MA (0.86 g, 3.2 mmol), AMA (50 mg, 0.3 mmol), and PMDETA (21 μL, 0.1 mmol) were mixed into a reaction flask. Then, the mixed solvents of 2-propanol (2 mL) and N′N-dimethylformamide (DMF, 2 mL) were added to dissolve the reaction mixture. The catalyst CuBr (14 mg, 0.1 mmol) was added into the flask under a nitrogen atmosphere after three cycles of freeze–pump–thaw to remove the dissolved oxygen, and the polymerisation was maintained at 40 °C for 12 hours. Next, the reaction mixture was diluted with THF (10 mL), and passed through a neutral aluminium oxide column to remove the catalyst CuBr. The residue was dialysed in Milli-Q water for 48 h and lyophilised finally. A series of purified copolymers were further characterised by 400 MHz $^1$H-NMR (MestReNova 9.0).

To measure the p$K_a$ levels, copolymers were dissolved in 10 M HCl at the final concentration of 10 mg/mL. Aliquot of 1 M NaOH solution (10 μL) was pipetted under stirring. The pH of the mixture was monitored with a Mettler Toledo pH meter. The pH value versus volume of NaOH solution was plotted in Supplementary Fig. 5a, and p$K_a$ values are listed in Supplementary Table 1.

For fluorescent labelling, each dye-NHS (10 mg mL$^{-1}$) and copolymer (50 mg mL$^{-1}$) were mixed in anhydrous DMF and stirred for 24 h in the dark. Then the copolymers were purified by preparative gel permeation chromatography to remove the residual small-molecule probes. The dye-conjugated copolymers were lyophilised and stored for further investigation.

### Syntheses and characterisation of PEG-$b$-P(R-$r$-GFLG-IMDQ) block copolymers
The polymer-drug conjugates PEG-P(DBA$_{48}$-$r$-D5A$_{32}$-$r$-GFLG-IMDQ) and PEG-P(EH$_{80}$-$r$-GFLG-IMDQ) were obtained via reversible addition-fragmentation chain transfer (RAFT) polymerisation method. Taking PGN$_{4.9}$ as an example, $PEG_{5k}$-CTA (100 mg, 0.02 mmol), DBA-MA (231 mg, 0.96 mmol), D5A-MA (172 mg, 0.64 mmol), MA-GFLG-IMDQ (160 mg, 0.2 mmol) and AIBN (0.82 mg, 5.0 μmol) were mixed into a reaction flask with anhydrous DMF, followed by three cycles of freeze–pump–thaw to remove oxygen. The polymerisation was kept under 65 °C for 48 h, and the reaction was successively dialysed in DMF and Milli-Q water for 48 h. Finally, the solution was lyophilised for storage at −20 °C. The purified copolymers were further characterized by 400 MHz $^1$H-NMR and gel permeation chromatography in Supplementary Table 2.

### Preparation and characterisation of PGNs
A series of binary ratiometric nanoreporters comprised of "OFF−ON" modules and "Always-ON" modules with specific molar ratios were

prepared according to our previous works[24] (BDP: Cy3.5 = 6: 4; Cy5: Cy7.5 = 9: 1). Briefly, the fluorescent copolymers were dissolved in methanol and added into Milli-Q water under sonication for 40 seconds. After methanol removal by micro-ultrafiltration tube (100 kDa, Merck Millipore) five times, the suspension was concentrated to 4 mg mL$^{-1}$ and stored at room temperature for subsequent studies. Meanwhile, PGN$_{4.9}$ and NPGN nanoadjuvants were prepared following a similar procedure. The hydrodynamic diameter and zeta potentials of nanoparticles (800 μg mL$^{-1}$) were measured by Zetasizer Nano ZSP (Malvern Instruments) at room temperature. The morphology at different states was obtained by transmission electron microscope (JEM-1400, JEOL).

### pH sensitivity and stability of PGN nanoreporter in vitro

A fluorescence spectrophotometer (F-7000, Hitachi) was used to assess the emission spectra of PGN nanoparticles. The stock solution was diluted to 100 μg mL$^{-1}$ in PBS buffer solutions with a series of pH values. BDP, Cy3.5, Cy5 and Cy7.5 (ICG) were excited at 488, 575, 635 and 780 nm, respectively. The peak intensity was processed to calculate the values of pH$_t$, R$_F$ ($F_{max}/F_{min}$) and ΔpH$_{10\%-90\%}$. Meanwhile, the IVIS Spectrum imaging system (Perkin Elmer) was also used to visualise fluorescence 'OFF−ON' and 'Always-ON' phenomena.

To evaluate the stability of nanoparticle, the stock solution was diluted to 100 μg mL$^{-1}$ with PBS of different pH (7.4 or 4.0) and fresh plasma, respectively. Incubating at 37 °C, the fluorescence signals of diluted solutions were measured at predesignated time points by the fluorescence spectrophotometer.

### Subcellular trafficking of PGN

For in vitro experiments, BMDMs were cultured in a glass-bottom dish and polarised to different phenotypes according to the aforementioned methods. PGN$_{4.9}$-BDP (100 μg mL$^{-1}$) was added to the dish and cocultured for 2 h. Lysosomes were labelled with LysoTracker Red. Intracellular trafficking of PGN was captured by a confocal microscope (A1R-Storm, Nikon) under a 100× oil objective lens.

For the subcellular trafficking studies of PGN in vivo, PGN-Cy5 nanoparticles (20 mg kg$^{-1}$) were intravenously injected into orthotopic 4T1 tumour-bearing mice. Tumours were dissected at 24 h post-injection and prepared as frozen sections by Cryostat (Leica CM1950). These tumour slides were fixed, exposed to anti-Lamp1 antibody (ab25245, clone number: 1D4B, Dilution 1:100) overnight at 4 °C, and then stained with DyLight 488-conjugated Goat anti-Rat secondary antibody (A23240, Dilution 1:1000) for 2 h. All the slides were mounted with Hoechst 33342-containing medium, fluorescently imaged by the confocal microscope (A1R-Storm, Nikon) under a 100× oil objective lens.

### Intracellular activation of PGN nanoreporter

For the CLSM experiment, all cell lines were pulsed with fluorescent nanoparticles (100 μg mL$^{-1}$) on the ice for 10 min, washed twice with ice-cold PBS, followed by a 4 h chase at 37 °C. After incubation, cell nuclei were stained with Hoechst 33342 (5 μg mL$^{-1}$) for 10 min. The fluorescence images were captured by a confocal laser scanning microscope (ZEISS LSM880) under a 63× oil objective lens. The Hoechst 33342, BDP, and Cy3.5 were excited at 405 nm, 488 nm, and 561 nm, respectively. The ratiometric images were visualised and quantified via ImageJ software (NIH). For flow cytometry analysis, BMDMs were cultured in 24-well plates and polarised to different phenotypes according to the aforementioned methods. After incubation with PGN-BDP/Cy3.5 (100 μg mL$^{-1}$) in DMEM medium at specified time points, BMDMs ($n = 3$) were washed with ice-cold twice and collected in cell staining buffer (BioLegend) for quantification by flow cytometer (FACSCalibur, BD, USA).

### In vivo and ex vivo fluorescence imaging

PGN-ICG or PGN-Cy5/Cy7.5 nanoparticles (20 mg kg$^{-1}$) were intravenously injected into orthotopic 4T1 tumour-bearing mice ($n = 4$ for each group). Then fluorescence images were captured by IVIS Spectrum imaging system (Perkin Elmer) at predesignated time-points ($\lambda_{ex}/\lambda_{em}$: 620 ± 10 nm/670 ± 20 nm for Cy5, 780 ± 10 nm/845 ± 20 nm for Cy7.5 and ICG). At 24 h post-administration, mice were perfused with PBS, and the dissected tumours and major organs were imaged fluorescently. The ratiometric images were processed and quantified by ImageJ software (NIH).

### Immunofluorescence staining

The excised 4T1-GFP tumours were cryosectioned and sliced into adjacent 10 μm slices at −20 °C by Cryostat (Leica CM1950). These tumour slides were fixed, exposed to anti-iNOS antibody (ab178945, clone number: EPR16635, Dilution 1:1000) or anti-mouse CD206 antibody (141702, Biolegend, clone number: C068C2, Dilution 1:200) overnight at 4 °C, respectively, and then stained with Alexa Flour 594-conjugated Goat anti-Rat secondary antibody (A23440, Dilution 1:1000) for 2 h. All the slides were finished with Hoechst 33342-containing mounting medium, fluorescently scanned and quantitatively analysed by the Vectra Polaris analysis system (Perkin Elmer).

### Flow cytometry analysis of cell population in tumours

To measure the percentage of M2-like TAMs, excised tumours from various tumour models were prepared for single-cell suspension after in vivo fluorescence imaging by the IVIS Spectrum imaging system. The samples were stained with PerCP/Cy5.5 anti-mouse CD45 antibody (103132, Biolegend, clone number: 30-F11, Dilution 1:100), FITC anti-mouse/human CD11b antibody (101205, Biolegend, clone number: M1/70, Dilution 1:200), PE/Cy7 anti-mouse F4/80 antibody (123114, Biolegend, clone number: BM8, Dilution 1:100), and PE anti-mouse CD206 antibody (141706, Biolegend, clone number: C068C2, Dilution 1:40) according to the manufacturer's instructions. The proportions of CD11b$^+$F4/80$^+$CD206$^+$ macrophages were obtained by flow cytometry, and the correlation analysis between M2-like macrophage content and the ratiometric signal was measured by GraphPad Prism 8 software.

### Quantification of PGN accumulation and activation efficiency

To quantify the tumour accumulation, a calibration curve was established. Tumour tissues were dissected from 4T1 tumour-bearing mice, grinded by homogeniser (T10 basic Ultra-Turrax) and mixed with three PGN nanoreporters at different concentrations (0, 0.1, 0.5, 1, 5, 10 and 50 μg mL$^{-1}$) in acid methanol plus 0.01% tritonX-100 (100 mg tumour in 500 μL solvent), respectively. After centrifugation at 13,800×$g$, a series of supernatants were imaged and quantified using the IVIS Spectrum imaging system ($\lambda_{ex}/\lambda_{em}$: 780 ± 10 nm/ 845 ± 20 nm). The linear regression analysis of fluorescence intensity versus nanoparticle concentrations was processed by GraphPad Prism 8 software.

For nanoparticle accumulation measurement, tumour homogenates were prepared from 4T1 tumour-bearing mice at 24 h post-injection of ICG-conjugated PGN and visualised by the aforementioned methods. The accumulation level of PGN nanoparticles was calculated using the prepared standard curves.

To quantify the activation efficiency of PGNs, the ICG signal of tumour tissue for each mouse at 24 h post-injection was divided by nanoparticle accumulation in the same tumour samples. The activation efficiency of different PGNs in tumour tissues was normalised to the counterpart of PGN$_{4.3}$ for parallel comparison. Meanwhile, the accumulation level and activation efficiency of PGNs in other major organs (e.g. livers and spleens) were also measured via the same procedure.

## Ex vivo fluorescence imaging of lymph nodes

PGN-ICG or PGN-Cy5 nanoparticles (20 mg kg$^{-1}$) were intravenously injected into orthotopic 4T1 tumour-bearing mice ($n = 5$ for each group). At 24 h post-administration, the dissected lymph nodes were fluorescently imaged and quantified by the IVIS Spectrum imaging system (Perkin Elmer, $\lambda_{ex}/\lambda_{em}$: 780 ± 10 nm/ 845 ± 20 nm). The slides of excised lymph nodes in the PGN-Cy5 group were sequentially stained with anti-mouse F4/80 antibody (ab6640, clone number: CI:A3-1, Dilution 1:500) and DyLight 488-conjugated Goat anti-Rat secondary antibody (A23240, Dilution 1:1000) according to the manufacturer's instructions.

## In vitro characterisation of AND-gated behaviour

The IMDQ release profile was evaluated using high-performance liquid chromatography (HPLC). PGN$_{4.9}$ nanoadjuvant was dispersed into different pH media (0.1 M Na$_2$HPO$_4$/citric acid buffer) with or without papain (30 mM) and incubated at 37 °C. At predesignated time points, the solution was taken out and quenched by adding acetonitrile. The amount of the released IMDQ was detected by HPLC at a UV wavelength of 322 nm. Chromatographic column: ZORBAX Eclipse Plus C8. Mobile phase: acetonitrile: 0.3% acetic acid = 10: 90–90: 10.

## TLR agonistic activity on TLR reporter cells

The reporter cell assay reveals TLR activation and downstream NF-κB signalling by secreting embryonic alkaline phosphatase[16]. The RAW-Blue macrophages were cultured in a 96-well plate with 50,000 cells/well in the complete heat-inactivated DMEM medium. After incubation with different IMDQ formulations at a series of concentrations for 24 h, 50 μL of supernatant was collected from each well and mixed with 150 μL colouring substrate Quanti-blue solution (InvivoGen). The absorbance was measured by a Microplate Reader (Multiskan FC, Thermo Fisher Scientific) at 620 nm. The TLR activation curves and the EC$_{50}$ values were calculated by GraphPad Prism 8 software.

## In vitro safety evaluation

RAW264.7 macrophages, MC38 tumour cells and 3T3 fibroblast cells (5000 cells/well) were seeded into 96-well plates and incubated with various IMDQ preparations at a series of concentrations in DMEM medium for 24 h, respectively. Then the cell viability was evaluated by MTT assay. Each well was replaced with MTT solution (0.5 mg mL$^{-1}$, Sigma) for 4 h, and DMSO was mixed to dissolve the generated formazan precipitation. The absorbance intensity was measured by the Microplate Reader at 540 nm, and the IC$_{50}$ values were quantitatively obtained using Origin 2020b.

## Acute immune-activation evaluation

PGN$_{4.9}$, PGN$_{6.3}$ nanoadjuvants, and IMDQ solution (equivalent to 2 mg kg$^{-1}$ IMDQ) were injected intravenously into healthy C57BL/6 mice. The peripheral blood samples were obtained at predesignated time-points, centrifuged 1000 × $g$ for 10 min to extract plasma, and stored for enzyme-linked immunosorbent assay (ELISA, Peprotech) quantification of typical serum cytokines, including IL-12 and IP-10.

## In vitro repolarization of BMDMs

BMDMs with different phenotypes were cultured for repolarization assay. M2-like macrophages were incubated with PBS, IMDQ, NPGN, PGN$_{4.9}$ nanoadjuvants (equivalent to 10 μM IMDQ) and PGN$_{4.9}$ control polymer (w/o IMDQ) in DMEM medium for 24 h. For cell morphology study, BMDMs were stained with AF488-wheat germ agglutinin (WGA, Invitrogen), Hoechst 33342 and TRITC Phalloidin (YEASEN) for cell membrane, nucleus and cytoskeleton labelling, respectively. The fluorescence images were visualised by a confocal laser scanning microscope (ZEISS LSM880) under a 63× oil objective lens.

To image the expression of M1 and M2 markers, BMDMs were stained with APC anti-mouse CD86 antibody (105012, BioLegend,

clone number: GL-1, Dilution 1:80), and PE anti-mouse CD206 antibody (141706, BioLegend, clone number: C068C2, Dilution 1:40) according to the manufacturer's instructions. The statistical results were evaluated by flow cytometry. The supernatant was collected for analysis of proinflammatory cytokines IL-12 using ELISA kits (Peprotech).

## Shotgun proteomics

BMDMs with different phenotypes and PGN$_{4.9}$ nanoadjuvant-treated M2-like macrophages were lysed in 0.5% sodium deoxycholate containing 25 mM Tris, 150 mM NaCl, 0.1% SDS, 1% Triton X-100, cocktail protease inhibitor and phosphatase inhibitor (Roche). Cell lysates were clarified by centrifugation at 13,800 × $g$ for 10 min and quantified to be 200 μg protein from each sample, followed by trypsin (Promega) digestion overnight for subsequent liquid chromatography–mass spectrometry analysis.

## Analysis of lysosomal protein expression

Western blot analysis and CLSM assay were used to assess the cellular protein content and activity of α-tubulin, iNOS, arginase and typical cathepsin family. BMDMs with various IMDQ treatments were lysed with RIPA lysis buffer containing protease and phosphatase inhibitors (Roche). The extracted protein was quantified via BCA Protein Assay Kit, fractionated on 10% SDS-PAGE gel electrophoresis and transferred to 0.45 μm polyvinyl difluoride membranes for 2 h. After being blocked with 5% defatted milk at room temperature for 1 h, the transferred membrane was stained with primary antibodies against mouse anti-α-tubulin (T5168, sigma, Dilution 1:10,000), rabbit anti-iNOS (ab178945, clone number: EPR16635, Dilution 1:1000), rabbit anti-liver arginase (ab133543, clone number: EPR6672(B), Dilution 1:2000), rabbit anti-cathepsin B (ab214428, clone number: EPR21033, Dilution 1:1000), rabbit anti-cathepsin L antibody (SAB4300959, sigma, Dilution 1:500) and rabbit anti-cathepsin S antibody (ab232740, Dilution 1:1000) overnight at 4 °C, and HRP-conjugated goat anti-mouse (ab6789, Dilution 1:10,000) and goat anti-rabbit (ab6721, Dilution 1:10,000) secondary antibodies for 2 h. Finally, the protein bands were processed via ECL chemiluminescence.

To evaluate the activation of cathepsin B, cells were incubated with Magic Red Cathepsin B kit (ImmunoChemistry) while protected from light for 15 min. After being washed twice by PBS, cell nuclei were stained with Hoechst 33342 (5 μg mL$^{-1}$) for 10 min. The fluorescence images were visualised with a confocal microscope (A1R-Storm, Nikon) under a 100× oil objective lens and processed by ImageJ software (NIH).

## DQ-OVA degradation assay

After incubation with various IMDQ formulations (equivalent to 10 μM IMDQ) for 24 h, BMDMs were incubated with DQ-OVA (10 μg mL$^{-1}$, Invitrogen) in heat-inactivated DMEM medium for 15 min, washed twice with PBS and incubated at 37 °C for another 15 min. DQ-OVA (488 nm) fluorescence was determined by confocal microscope (A1R-Storm, Nikon) and flow cytometry to assess the lysosomal degradative capacity of macrophages with different phenotypes.

## The expression of MHC-I molecule on BMDMs

Pretreated BMDMs were harvested and resuspended in staining buffer, followed by staining with PE anti-mouse H-2K$^d$ antibody (116608, Biolegend, clone number: SF1-1.1, Dilution 1:40). Mean fluorescence intensity of MHC-I molecules on the surface of cell membranes in macrophages was evaluated by flow cytometry.

## In vitro antigen presentation of BMDMs

BMDMs were treated with various IMDQ formulations, and OVA$_{257-264}$ (SIINFEKL, 10 μg mL$^{-1}$, Sangon Biotech) or OVA$_{257-280}$ (SIINFEKL-TEWTSSNVMEERKIKV, 10 μg mL$^{-1}$, Biomatik) for 24 h. Antigen-presenting cells were collected for staining of costimulatory factors,

including PE-Cy7 anti-mouse CD80 antibody (104734, Biolegend, clone number: 16-10A1, Dilution 1:40), APC anti-mouse CD86 antibody (105012, Biolegend, clone number: GL-1, Dilution 1:80), as well as PE anti-mouse SIINFEKL-H-2K$^b$ antibody (116608, Biolegend, clone number: SF1-1.1, Dilution 1:40) to evaluate the antigen-processing capacity of BMDMs by flow cytometry.

## Co-transplantation of BMDMs and tumour cells

M2-like BMDMs were pretreated with vehicle, IMDQ and PGN$_{4.9}$ nanoadjuvant (equivalent to 2 mg kg$^{-1}$) for 24 h. The repolarized M2-like macrophages and 4T1 cells ($1 \times 10^6$) were co-inoculated subcutaneously into the right flanks of BALB/c mice at a ratio of 1: 1. Tumour growth curves were monitored every other day with electronic callipers. Tumour volumes were calculated using the following equation:

$$\text{Tumour volume} = (\text{length} \times \text{width} \times \text{width})/2 \qquad (1)$$

## Depletion of macrophages in vivo

To investigate the effect of macrophages on nanoparticle activation, the tumour macrophages were depleted by intravenous injection of clophosome (FormuMax, 200 µL/mouse) every five days for three times. PGN$_{4.9}$-Cy5/Cy7.5 (20 mg kg$^{-1}$) was intravenously administrated 48 h after the last treatment with clophosome. The fluorescence images were visualised by IVIS Spectrum imaging system (Perkin Elmer) with time-lapse by the above-mentioned methods. Meanwhile, the depletion of F4/80$^+$CD206$^+$ TAMs was confirmed using flow cytometry.

To elucidate the mechanism of reprogramming therapy, mice were intraperitoneally injected with anti-mouse CSF1R (BE0213, 300 µg/mouse) every 5 days twice, followed by PGN$_{4.9}$ nanoadjuvant (equivalent to 2 mg kg$^{-1}$ IMDQ) administration at 24 h after macrophage blocking. Cellular depletion of systemic macrophages in peripheral blood mononuclear cells was demonstrated by flow cytometry. Tumour growth curves were monitored every other day.

## Antigen presentation of macrophages in vivo

MC38.OVA tumour-bearing mice were injected intravenously with PBS, IMDQ, NPGN and PGN$_{4.9}$ nanoadjuvants (equivalent to 2 mg kg$^{-1}$ IMDQ) every four days for three times. Four days after the last treatment, popliteal lymph nodes and inguinal lymph nodes were dissected and prepared into a single-cell suspension. The cells were stained with Brilliant Violet 510 anti-mouse F4/80 (123135, Biolegend, clone number: BM8, Dilution 1:40), PE-Cy7 anti-mouse CD80 antibody (104734, Biolegend, clone number: 16-10A1, Dilution 1:40), APC anti-mouse CD86 antibody (105012, Biolegend, clone number: GL-1, Dilution 1:80), and PE anti-mouse SIINFEKL-H-2K$^b$ antibody (116608, Biolegend, clone number: SF1-1.1, Dilution 1:40) and detected by flow cytometry. Meanwhile, tumour tissues were harvested and digested by a tumour dissociation kit (Miltenyi Biotec). The single-cell suspension in different groups was stained with PerCP/Cy5.5 anti-mouse CD45 (103132, Biolegend, clone number: 30-F11, Dilution 1:100), FITC anti-mouse/human CD11b (101205, Biolegend, clone number: M1/70, Dilution 1:200), Brilliant Violet 510 anti-mouse F4/80 (123135, Biolegend, clone number: BM8, Dilution 1:40), and PE anti-mouse SIINFEKL-H-2K$^b$ antibody (116608, Biolegend, clone number: SF1-1.1, Dilution 1:40). SIINFEKL$^+$ cells in CD11b$^+$F4/80$^+$ macrophages were measured by flow cytometry.

## In vivo specific killing cell assay

MC38.OVA tumour-bearing mice were intravenously injected with PBS, IMDQ, NPGN and PGN$_{4.9}$ nanoadjuvants (equivalent to 2 mg kg$^{-1}$ IMDQ) at 5, 9 and 13 d. At 15 d, spleens from naïve mice were processed to single-cell suspension. The splenocytes were pulsed with OVA$_{257-264}$ (10 µg/mL) or blank DMEM medium for 30 min, and labelled with 5 µM

or 0.5 µM carboxyfluorescein succinimidyl ester (CFSE, BD) for 15 min, respectively. After staining, CFSE$^{high}$ and CFSE$^{low}$ splenocytes were equally mixed ($1 \times 10^7$) and injected into treated mice via the tail vein. After two days, immunised mice were euthanized, and the CFSE signal of the splenocytes was measured by flow cytometry analysis. The specific killing percentage is calculated according to the following formula:

$$\text{Percentage of specific killing} = \left[ 1 - \frac{\text{CFSE}^{high}/\text{CFSE}^{low}}{\text{CFSE}^{high}(\text{PBS})/\text{CFSE}^{low}(\text{PBS})} \right] \times 100\%$$

$$(2)$$

## Cytotoxic T-cell activation

MC38.OVA tumour-bearing mice were immunised with different IMDQ formulations (equivalent to 2 mg kg$^{-1}$ IMDQ) every 4 days three times. The single-cell splenocytes were cultured with a cell stimulation cocktail (plus protein transport inhibitors) for 6 h at 37 °C. The cells were collected and stained with APC anti-mouse CD8α antibody (100712, Biolegend, clone number: 53-6.7, Dilution 1:80) and PE anti-mouse IFN-γ antibody (505808, Biolegend, clone number: XMG1.2, Dilution 1:100) according to the manufacturer's instructions. The percentages of CD8α$^+$IFN-γ$^+$ activated T cells were quantified by flow cytometer.

## Long-term memory T lymphocytes activation

To investigate the mechanism by which PGN$_{4.9}$ nanoadjuvant led to sustained tumour regression, MC38 tumour-bearing mice were injected with PBS, IMDQ, NPGN and PGN$_{4.9}$ nanoadjuvants (equivalent to 2 mg kg$^{-1}$ IMDQ) at day 0, 4 and 8. Two weeks after the last treatment, spleens of the treated mice were harvested and stained with FITC anti-mouse CD3 antibody (100204, Biolegend, clone number: 17A2, Dilution 1:50), PE anti-mouse CD4 antibody (100512, Biolegend, clone number: RM4-5, Dilution 1:80), APC anti-mouse CD8α antibody (100712, Biolegend, clone number: 53-6.7, Dilution 1:80), PE/Cy7 anti-mouse CD44 antibody (103030, Biolegend, clone number: IM7, Dilution 1:80), and Pacific Blue anti-mouse CD62L antibody (104424, Biolegend, clone number: MEL-14, Dilution 1:200). CD44$^{high}$CD62L$^{low}$ effector memory T cells and CD44$^{high}$CD62L$^{high}$ central memory T cells in CD8$^+$ and CD4$^+$ T cell subsets were measured by flow cytometer[47].

## In vivo tumour regression and rechallenge

4T1-luc breast cancer cells ($2 \times 10^5$) and MC38 colorectal cancer cells ($2 \times 10^6$) were implanted subcutaneously in BALB/c or C57BL/6 mice, respectively. The groups of mice were immunised by intravenous injection of PBS, IMDQ, NPGN and PGN$_{4.9}$ nanoadjuvants (equivalent to 2 mg kg$^{-1}$ IMDQ) on days 3, 7 and 11 post-tumour inoculation. Taking the 4T1-luc orthotopic tumour model as an example, tumour volumes were measured every other day, and the bioluminescence intensity of 4T1-luc cells was monitored by the IVIS Spectrum imaging system (Perkin Elmer). When tumour volume reached greater than 1500 mm$^3$, tumour tissues were collected, weighed and imaged. Tumour sections were processed with multi-colour immunohistochemistry staining for immune microenvironment analysis. Also, ex vivo lungs were collected for bioluminescence imaging to monitor lung metastasis, and lung sections were prepared for H&E staining on day 28. For MC38 colorectal tumour model, tumour-free mice of the PGN$_{4.9}$ nanoadjuvant group ($n = 6$ mice) were rechallenged with the second and third injection of MC38 cells ($2 \times 10^6$) on 73 days and 120 days post tumour inoculation, respectively. For combination treatment with PGN$_{4.9}$ nanoadjuvant and PDPA-DTX, 4T1-bearing mice were intravenously injected with PBS, PGN$_{4.9}$ nanoadjuvant (equivalent to 1 mg kg$^{-1}$ IMDQ) and PDPA-DTX (equivalent to 3 mg kg$^{-1}$ DTX), alone or in combination on day 7, 11, 15 and 19 post 4T1 tumour inoculation. The tumour volumes were measured by electronic callipers every other day, and

the survival curves were recorded according to Kaplan-Meier analysis. Furthermore, tumour tissues were harvested 5 days after the last treatment and digested by a tumour dissociation kit (Miltenyi Biotec). The single-cell suspension in different groups was stained with PerCP/Cy5.5 anti-mouse CD45 (103132, Biolegend, clone number: 30-F11, Dilution 1:100), FITC anti-mouse CD3 antibody (100204, Biolegend, clone number: 17A2, Dilution 1:50), PE anti-mouse CD4 antibody (100512, Biolegend, clone number: RM4-5, Dilution 1:80), and APC anti-mouse CD8α antibody (100712, Biolegend, clone number: 53-6.7, Dilution 1:80). $CD3^+CD8^+$ and $CD3^+CD4^+$ T lymphocytes were measured by flow cytometry.

### Depletion of T lymphocytes in vivo

The 4T1 tumour-bearing mice were intraperitoneally injected with anti-mouse CD8 (BE0117, 400 μg/mouse) every 3 days for three times, followed by $PGN_{4.9}$ nanoadjuvant (equivalent to $2\,mg\,kg^{-1}$ IMDQ) administration at 24 h after T cell blocking. Depletion of $CD3^+CD8^+$ T lymphocytes in peripheral blood mononuclear cells was demonstrated by flow cytometry, while tumour growth curves were monitored every other day ($n = 7$ mice for $PGN_{4.9}$ groups; $n = 9$ mice for other groups).

### Combined therapeutic efficacy of $PGN_{4.9}$ nanoadjuvant plus α-PD1

MC38 tumour-bearing mice were randomly divided into four groups and treated with PBS, α-PD1, $PGN_{4.9}$ nanoadjuvant, and α-PD1 + $PGN_{4.9}$ nanoadjuvant. $PGN_{4.9}$ nanoadjuvant was intravenously injected into the C57BL/6 mice (equivalent to $2\,mg\,kg^{-1}$ IMDQ) on days 3, 9 and 15. Simultaneously, mouse α-PD1 (100 μg/mouse, BioXcell) was administered intraperitoneally on days 0, 3, 6 and 9 ($n = 11$ mice for the PBS group; $n = 10$ mice for other groups). The tumour volumes were measured by electronic calipers every 2 days, and the survival curves were recorded according to Kaplan–Meier analysis. Besides, the tumour samples were stained with PerCP/Cy5.5 anti-mouse CD45 (103132, Biolegend, clone number: 30-F11, Dilution 1:100), FITC anti-mouse CD3 antibody (100204, Biolegend, clone number: 17A2, Dilution 1:50), PE anti-mouse CD4 antibody (100512, Biolegend, clone number: RM4-5, Dilution 1:80), APC anti-mouse CD8α antibody (100712, Biolegend, clone number: 53-6.7, Dilution 1:80), and BV421 anti-mouse CD25 (102043, Biolegend, clone number: PC61, Dilution 1:200) according to the manufacturer's instructions. The proportions of $CD3^+CD8^+$ cytotoxic T lymphocytes and $CD4^+CD25^+$ regulatory T lymphocytes were obtained by flow cytometry.

### Statistics and reproducibility

Confocal imaging of macrophages with or without adjuvant treatments incubated with binary nanoreporter, the expression of cathepsin B and DQ-OVA degradation assays were repeated at least three times with similar results, and a series of representative images from each group were shown, such as Figs. 2d, e and 4f, i, k. For whole-mount images of tumour adjacent slices from 4T1-GFP tumour-bearing mice at 24 h post-administration of PGNs, the experiment was repeated thrice with similar results; the representative images were shown in Fig. 3f. The results of western blots were repeated thrice independently with similar results, and one representative image from each group was shown in Fig. 4e, j.

### Statistical analysis

Data were presented as mean ± s.d. and analysed by GraphPad Prism 8 and Origin 2020b. The significant differences among treatment groups were analysed using student's $t$-tests or one-way (or two-way) ANOVA followed by Tukey's multiple comparisons test. Welch's correction was applied for the groups without equal s.d. Long-term survival curves were compared by log-rank test, and $P < 0.05$ was regarded as statistically significant.

### Reporting summary

Further information on research design is available in the Nature Portfolio Reporting Summary linked to this article.

## Data availability

The mass spectrometry proteomics data have been deposited to the ProteomeXchange Consortium via the PRIDE partner repository with the dataset identifier PXD044911. The remaining data are available within the Article, Supplementary Information or Source Data file. Source data are provided in this paper.

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

## Acknowledgements

This work was supported by the National Natural Science Foundation of China (NSFC) grants (82225044 and 81973260 to Y.W.) and the National Key Research and Development Programme of China (2021YFA1201200 to Y.W.). We would like to thank the State Key Laboratory of Natural and Biomimetic Drugs, Peking University Biological Imaging and Flow Cytometry Core Facilities for flow cytometry, confocal, animal, and tis-sue imaging services.

## Author contributions

M.T. and Y.W. are responsive for all phases of this research. R.Z. and H.X. helped with the synthesis of fluorescent copolymers and drug-conjugated copolymers. B.C., H.X. and J.Z. helped with the immunolo-gical experiments. M.P., Q.Y. and C.F. participated in the cell and animal imaging studies. F.W. and Y.Y. helped with the western blot experi-ments. L.Z. performed the shotgun proteomics analysis. M.T. and Y.W. wrote the manuscript. B.C., Q.Z. and Y.W. provided the conceptual advice and supervised the study. All authors discussed and commented on the results.

## Competing interests

The authors declare no competing interests.
