## [Peer Review File · Nature Communications]

pH-gated nanoparticles selectively regulate lysosomal function
of tumour-associated macrophages for cancer
immunotherapyREVIEWER COMMENTS

Reviewer #1 (Remarks to the Author): with expertise in nanomedicine, cancer immunology

In the manuscript authors developed a pH-gated nanoparticle (PGN) adjuvant that selectively targets the lysosomes of M2-like TAMs in tumours. The authors show convincing data that they were able to design a pH-gated nanoparticle, PGN4.9, that only opens in M2 macrophages both in vitro and in vivo. In addition, using PGN4.9-delivered imidazoquinoline (IMDQ), they were able to differentiate M2 into M1 macrophages in the tumor microenvironment. PGN4.9 IMDQ itself lead to a reduced tumor growth but could also be combined with other immune therapies to further decrease tumor growth.

The findings are novel, original and well-supported by the data.

Specific comments:

One issue that should be addressed is that the text is written in an overcomplicated manner which makes it difficult to understand some sections (example: lines 84-96). Secondly, in this manuscript the authors designed a delivery method for an immunomodulatory drug. The term “nanoparticle adjuvant” or “PGN adjuvant” frequently used in the manuscript is misleading, since the adjuvant is the nanoparticle drug cargo and not the particle itself.

Figure 2, please show that the particles are taken up specifically in the lysosome (for instance by costaining). 2D) Please choose cells with the same amount of particles (Cy3.5) inside the cell.

Fig 3, Please also show lysosomal uptake in vivo.

3D) please discuss why PGN4.3 and PGN4.9 accumulate but only PGN4.9 and PGN6.3 activate.

3E) Please discuss why the particles don't open in the other organs harboring M2 macrophages (liver/spleen). In this figure no controls, or other particles, are shown. Please also show this data for the other particles.

3F) For clarity, please use the same colors between pie chart and confocal pictures.

3J) the word 'dramatic' is exaggerated

4B) please call it a reporter assay in the text rather than RAW blue. In my understanding, for the functionality of the particle, the lysosomal pH is very important. If so, please show the pH of the RAW lysosome.

SI21)a, m1 and m2 are the untreated cells? And the particles are tested on which type of macrophages? please clarify.

SI22) Please explain why the H&E staining shows good bio compatibility

5A/B/C) Please also show the raw count rather than only percentages

SI32D) Please discuss why there are equal amounts of M2 macrophages between the IMDQ, NPGN and PGN4.9.

5G) Please explain how the CFSE dye correlates with killing.

6I/J) As the PDPA-DTX is not an established treatment that increases antigen presentation, please show this. It would also be interesting to see immune infiltration data for this experiment to determine how it worked.

6K) Why the sudden switch from 4T1 to MC38 in this figure? All previous experiments were only demonstrated for 4T1. Did they also work with MC38?

6L/M/N) The figures show a combination therapy using anti-PD1 and PGN4.9 IMDQ.

However, only tumor growth kinetics and survival are shown. Please also show infiltration data to indicate how the combination therapy might work.

Literature describes that activating existing M1 macrophages using imidazoquinoline TLR ligands can inhibit tumor growth. The authors should discuss why only targeting M2 macrophages would be more beneficial rather than targeting both type of macrophages.

The work has not been put into perspective in the final section of the manuscript. It is important that the authors show the relevance in comparison to current literature on this topic.

Reviewer #2 (Remarks to the Author): with expertise in tumor associated macrophages

Tang et al present pH-gated nanoparticles, termed PGN4.9, that degrade in the lysosomal

compartment of M2-oriented macrophages, but not M0/M1 or other normal cell types or cancer cells. The particles accumulate in the tumor microenvironment and, when loaded with the TLR7/8 ligand IMDQ, are able to skew the macrophage activation state to M1, with a higher antigen-presenting capacity and anti-tumor activities. Consequently, PGN4.9 collaborates with anti-PD1 to increase anti-tumor immunity. This is an interesting story, but some remarks remain:

- 1) I don't understand why PGN6.3 would accumulate less in the tumor. Accumulation is influenced by size, stability etc, so is this different for PGN6.3?
- 2) The authors conclude that PGN4.9 does not get activated in normal tissues, based on observations in the liver and the spleen. However, the authors should look at organs where more M2-oriented macrophages are likely to reside, such as eg the lung and possibly also the skin
- 3) Fig 3g shows a correlation between the ratiometric signal and the presence of CD206+ TAMs. However, tumors may also contain M1-like TAMs. How is the correlation with those TAMs? The authors should more directly demonstrate that PGN4.9 fluoresces only within the more M2-oriented TAMs
- 4) Fig 4b also shows that PGN4.9 adjuvant (I presume that this is the PGN4.9 particle conjugated to IMDQ, although I don't find this very clear from the text) stimulates TLR a lot less than free IMDQ. Please explain.
- 5) The experimental conditions used in Figure 4 should be clearer represented on the figures or in the legend. If the authors use "PGN4.9", what does that mean? I presume they mean M2 + PGN4.9, but that's nowhere stated. Could also be M0 + PGN4.9.
- 6) Along the same line, What are the conditions in Figure 4f? I guess "M2" means a treatment with PGN4.9+OFF/ON fluorescent module, while "PGN4.9" means PGN4.9+IMDQ? I can only guess...
- 7) Fig 4m. Are the MHC-I/SIINFEKL complexes higher in the PGN4.9 condition because more SIINFEKL peptide is produced or because there is a higher overall surface expression of MHC-I molecules? The authors should discriminate between these possibilities by assessing the production of SIINFEKL in M2 versus M2+PGN4.9
- 8) The authors investigated the biodistribution of the particles in the tumor, but not the tumor-draining lymph node. In Fig 5, however, they suddenly look at the tdLN macrophages instead of the TAMs. What happens to the TAMs in terms of antigen processing and

presentation?

9) Fig 5h. Why does NPGN result in a higher killing capacity than IMDQ?

10) Fig 6h. This means that these TAMs contribute to anti-tumor immunity. Do these TAMs kill cancer cells?

In addition, to what extent are CD8+ T cells important? This can be shown by in vivo depletion of these cells.

11) Fig 6k. The control group says "PBS", while these are simply control C57BL/6 mice.

Reviewer #3 (Remarks to the Author): with expertise in nanoparticles

In this manuscript, a pH-gated nanoparticles library was established for selectively target the highly lysosomal acidity of M2-like macrophages. The PGN4.9 nanoadjuvant with AND-gate capacity can selectively fine-tune lysosomal proteolysis of M2-like TAMs to facilitate antigen cross-presentation and provoke adaptive tumour immunity. The study is interesting, but some results are not very persuasive. Based on these points, I suggest that this manuscript can be accepted for publication after addressing following issues.

1. Line 437, the authors adopted fluorescence intensity to evaluate the PGN accumulation and activation efficiency. This method of quantitative analysis is not accurate. The PNG with the fluorescent label attached may have residual fluorescent label, or the fluorescent label may be free from the polymer. To assess the accumulation or metabolism of PNG, a more accurate method is still chromatography or chromatography-tandem mass spectrometry.
2. After PGN4.9 was injected into animals through the tail vein, how does it target the M2-like TAM in tumors? What is the mechanism of this targeting function? How was it considered when designing the synthesis?
3. By using what kind of method to verify the lysosomal pH value in BMDMs to obtain the fitting standard curve? Can you provide more experiment protocols?
4. The manuscript contains too many abbreviations, which makes the reading process not very smooth. The meaning of the acronym is often forgotten. If the phrase or word composition is not complex, try to use the original expression to make the article reading more smoothly. If a phrase occurs only once or twice, it is not necessary to assign a separate abbreviation. For example, TME stands for tumor microenvironment, APCs for antigen-presenting cells, and GPC for chromatography. It is not necessary to create an abbreviation

for these phrases or words.

5. In supplementary Fig. 5, in the plot, the author should elaborate what is the F instead for. End explain the Two coordinate system in accuracy, avoiding ambiguous understanding.

What F/Fmin instead for?

6. In supplementary Fig. 5, the fluorescence intensity of Cy3.5 is stable, while the BDP is fluctuate dramatically. Why?

7. In Line 422, “tumors were frozen and cut into several adjacent slices with 10 μ m thickness”. Do you mean “Tumor tissues were cryo-sectioned and sliced into 10-micron slices.” The type of instrument that used, and the freezing temperature should be described.

8. The design, grouping and experimental cycle of animal experiments are not well described. How many animals in each group need to be maintained for 150 days? During this period, in addition to the change and control of tumor volume, how can animal welfare be guaranteed? Is there a humanitarian end? Are other indicators of animals tracked and monitored? Are there any abnormalities in behavior, physiological and biochemical indicators?

9. A thoroughly check should be made to improve the English and correct the grammar and typo.

Reviewer #4 (Remarks to the Author): with expertise in nanomedicine, cancer immunology

This manuscript by Tang et al, presents an innovative approach consisting in the use of pH-gated nanoparticles with ability to deliver a drug (imiquimod) and regulate lysosomal function in tumor associated macrophages. The hypothesis, performance of experiments and results are appropriated, followed by a good presentation (both writing and figures). As the key element of this research, the development of pH-gated nanoparticles and their performance is very interesting and providing very satisfactory results, thus I recommend this manuscript for publication with minor delay after addressing the following minor concerns:

1 - Supp fig 6 legend is wrong. b) and C) have to be changed.

2 - supp fig 13b) requires more details for the performance of the readout. 13c) legend requires description of organs collected. It is not clear if the 4T1 model is s.c. or orthotopic (probably s.c.). These details should be clearly mentioned in the figure legend.

3 -Dramatic decrease of signal is too exaggerated for figure 3.h and j. or supp. fig. 17. Some signal is still visible in tumors after depleting macrophages. I miss the analysis of macrophage depletion? which percentage of macrophages was depleted? only M2 or also M1 were depleted? For how long?

4 -In supp. fig. 21 e-h, injected dose of IMDQ and others is missing.

5 -In supp. fig. 27, indication of experimental doses and times would be appreciated.

6 -In supp. fig. 33, dose of each treatment would be appreciated.

7 -In fig. 6, doses of treatments would be appreciated.

8 -In supp. fig. 36, time of evaluation must be indicated.

9 -In line 254, the sentence "The effects of PGN4.9 on tumour inhibition was mitigated obviously in macrophages-depleted mice" must be reformulated, because the "mitigation" is not so obvious indicating that other cell types are also important.. It could be also mentioned that macrophage depletion is not complete with the CSF1R antibody.

10 -For supp. fig. 38 and fig. 6i, the timing and doses of each therapeutic approach are not clear. These should be indicated.

11 -Discussion must be significantly improved according with the following comments. a)In the first paragraph of discussion more references are needed. b) More details and references about the nanotechnology would be appreciated (other pH sensitive NPs). c) drug delivery approach to target macrophages in tumors require appropriate discussion.

Point-by-point response to reviewers

We would like to thank the reviewers for the insightful and constructive comments! We have revised the manuscript according to their advices, which should significantly improve the clarity and quality of our work. Below is a list of the point-by-point responses to the reviewer comments shown in *italics* and the corresponding changes that we made highlighted in **yellow**.

Reviewer #1 (Remarks to the Author): with expertise in nanomedicine, cancer immunology

In the manuscript authors developed a pH-gated nanoparticle (PGN) adjuvant that selectively targets the lysosomes of M2-like TAMs in tumours. The authors show convincing data that they were able to design a pH-gated nanoparticle, PGN4.9, that only opens in M2 macrophages both in vitro and in vivo. In addition, using PGN4.9-delivered imidazoquinoline (IMDQ), they were able to differentiate M2 into M1 macrophages in the tumor microenvironment. PGN4.9 IMDQ itself lead to a reduced tumor growth but could also be combined with other immune therapies to further decrease tumor growth. The findings are novel, original and well-supported by the data.

Specific comments:

One issue that should be addressed is that the text is written in an overcomplicated manner which makes it difficult to understand some sections (example: lines 84-96). Secondly, in this manuscript the authors designed a delivery method for an immunomodulatory drug. The term “nanoparticle adjuvant” or “PGN adjuvant” frequently used in the manuscript is misleading, since the adjuvant is the nanoparticle drug cargo and not the particle itself.

We appreciate the Reviewer’s encouraging comments on the novelty and impact of the PGN nanotechnology. We have revised the manuscript according to your advices and other referee’s comments.

We have rewritten the mentioned sections to help understand the nanoparticle characterization experiment. Besides, we have corrected the original expression and made some changes in the revised manuscript. These changes will not influence the content and framework of the paper.

Lines 85-89: Furthermore, the pH_t of PGNs was determined to be within a range from 4.46 to 5.47 with sharp response ($\Delta pH_{ON/OFF} \sim 0.2-0.3$, Fig. 2b) using fluorescence analysis procedure²¹. To evaluate the pH-gating capacity of PGNs in living cells, we successfully constructed binary ratiometric nanoreporters, which consisted of the ‘OFF-ON’ and ‘Always-ON’ fluorescence modules in visible-light window²⁴.

1. *Figure 2, please show that the particles are taken up specifically in the lysosome (for instance by co-staining). 2D) Please choose cells with the same amount of particles (Cy3.5) inside the cell.*

Re: Per the reviewer’s advice, we have conducted the intracellular trafficking experiments on BMDMs co-staining with LysoTracker red, which is an indicator of lysosome. As shown in **Supplementary Fig. 7**, the yellow signals indicated the colocalization of PGN4.9 with lysosomes in BMDMs with different phenotypes.

We added the above results in **Supplementary methods** and **Supplementary Figure 7**.

Lines 99-101: We firstly imaged the intracellular trafficking of PGNs after incubation with living cells for 2 h. Results showed that PGNs had a good colocalization with lysosomes (Supplementary Fig. 7).

Supplementary Fig. 7. Intracellular distribution of PGN_{4.9} nanoprobe on M1- and M2-like BMDMs after incubation for 2 h. Scale bar, 10 μ m.

Per the reviewer's suggestion, we have replaced the original images with new representative confocal images for M1- and M2-like macrophages in **Figure 2d**. As for the M0-like macrophages, the lysosome number is pretty low due to its poor phagocytic activity. Thus, only small amount of Cy3.5-positive organelles were captured.

Figure 2. (d) Confocal images of macrophages with different phenotypes treated with PGN_{4.9}-BDP/Cy3.5 binary nanoreporter for 2 h. Ratiometric images represent the pH-gated activation of nanoparticle independent of its concentration. Green, BDP; red, Cy3.5; blue, nucleus; Scale bar, 10 μ m.

2. *Fig 3, Please also show lysosomal uptake in vivo.*

Re: We acknowledge the reviewer's kind suggestion. In order to investigate the distribution of nanoparticles *in vivo*, we performed immunofluorescence staining of Lamp 1 in tumour tissue sections to image the lysosome trafficking of PGN_{4.9}. As shown in **Supplementary Fig. 13**, we observed that a majority of nanoprobe were trapped in lysosomes at 24 h post-administration, as indicated by the good colocalization with Lamp 1.

We added the above results in **Supplementary methods** and **Supplementary Figure 13**.

Lines 124-128: To evaluate the lysosomal accumulation of PGNs *in vivo*, 4T1-tumour bearing mice

were intravenously injected with Cy5-conjugated PGN_{4.9}, and tumours were excised at 24 h post-injection and cryosectioned for the immunofluorescence staining of LAMP 1 (lysosome biomarker) in tumour slices. Results demonstrated that PGNs had a good colocalization with lysosomes within tumour tissues at 24 h post-administration (Supplementary Fig. 13).

Supplementary Fig. 13. Confocal images of endocytic organelles distribution of PGN_{4.9} nanoprobe in 4T1 tumour-bearing mice. Immunofluorescence staining of Lamp 1 (Green) was performed to label lysosomes. Scale bar, 10 μ m.

3. 3D) please discuss why PGN_{4.3} and PGN_{4.9} accumulate but only PGN_{4.9} and PGN_{6.3} activate.

Re: According to our previous studies (Li *et al*, *Nanomedicine* **2019**, 17: 287-296), All three ICG-conjugated micelles had long blood circulation with half-lives of more than 10 h, while the p*H*_t of nanoparticles have significant effect on their blood AUC_{0-24 h}. Moreover, our group performed an in-depth research on the effect of hydrophobicity of micellar core on its pharmacokinetics behavior, finding that lower p*H*_t of nanoparticles would lead to longer circulation time in vivo, as shown in **Figure R1**. Therefore, the tumour accumulation in PGN_{4.9}- and PGN_{4.3}-treated groups was 1.93- and 2.06-fold of that in PGN_{6.3}-treated group due to the higher blood AUC for PGNs with lower p*H*_t. On the other hand, we determined that the lysosomal p*H* of different cell types within tumour microenvironment is higher than 4.4 (p*H*_L > 4.4). Thus, PGN_{4.3} is hard to respond to lysosomal p*H* of M2-like TAMs, resulting in a dim fluorescence signal, whereas PGN_{6.3} and PGN_{4.9} can respond to the acidic signals within lysosomes and caused complete signal activation for lighting up tumour. Take these two factors into account, PGN_{4.9} can achieve both higher tumour accumulation and intracellular activation as compared with the other two PGNs.

Figure R1. Pharmacokinetics profiles of five ICG-conjugated nanoparticles with different p*H*_t (5.25-6.75) in tumour-free BALB/c mice (*n* = 3).

4. 3E) Please discuss why the particles don't open in the other organs harboring M2 macrophages (liver/spleen). In this figure no controls, or other particles, are shown. Please also show this data for the other particles.

Re: Our previous work has demonstrated that although the nanoparticles accumulated a lot in mononuclear phagocyte system (e.g., liver and spleen) over 24 h post-administration, the internalized amounts of nanoparticles only account for a very low percentage (3-5%) due to its limited endocytosis efficiency by cells (macrophage and hepatocytes) within liver and spleen. In contrast, about 20% of tumour accumulated nanoparticles were internalized by cells within tumour tissues (Yin *et al*, *Nat Commun* **2021**, 12,

2385). As shown in **Figure R2**, the most internalized NPs were not colocalized with macrophages in liver, which suggested that nanoparticles cannot efficiently endocytosed by macrophages in liver and spleen, unlike the behavior in tumour tissues.

Figure R2. The fluorescent images of liver slices from 4T1 tumour-bearing mice at 24 h post-administration of Cy5-labelled PGNs. Scale bar, 500 μ m. The macrophages were stained with anti-F4/80 antibodies.

Per the reviewer's advice, we have carried out new experiments about the accumulation level and activation efficiency in PGN_{6.3} and PGN_{4.3} control groups. The readouts of PGN_{6.3} and PGN_{4.3} activation efficiency in livers and spleens with macrophages were still significantly lower than that in tumour tissues.

We added the above results in **Supplementary methods** and **Supplementary Figure 17**.

Supplementary Fig. 17. Accumulation level and activation efficiency of (a) PGN_{6.3} and (b) PGN_{4.3} in dissected livers, spleens and tumours ($n = 4$). Activation efficiency in different groups was obtained by normalizing to that of PGN_{4.3} in tumour tissues.

5. 3F) For clarity, please use the same colors between pie chart and confocal pictures.

Re: Per the reviewer's suggestion, we have revised the color coding across panels in **Figure 3f** in the revised manuscript.

6. 3J) the word 'dramatic' is exaggerated

Re: Thanks for the suggestion. We have revised this sentence according to the statistical analysis.

Line 158-160: Moreover, depletion of 58.8% M2-like TAMs with clophosome caused a significantly decreased signal activation of PGN_{4.9}, further elucidated the selective targeting of PGN_{4.9} to M2-like TAMs (Fig. 3h-j; Supplementary Fig. 20).

7. 4B) please call it a reporter assay in the text rather than RAW blue. In my understanding, for the functionality of the particle, the lysosomal pH is very important. If so, please show the pH of the RAW lysosome.

Re: We appreciate the reviewer's advice and have changed the manuscript on RAW blue. We have measured the pH value of RAW-Blue with or without cytokine IL-4 by the ratiometric pH quantification, as shown in **Figure R3**. The results showed that the lysosomes of RAW-Blue with IL-4 ($pH_L \sim 4.50$) were hyper-acidified as compared with untreated RAW-Blue ($pH_L \sim 5.21$).

Figure R3. Lysosomal pH measurement of RAW-Blue. (a) Ratiometric images of different cell lines in buffer solutions with different pH and the corresponding cells in culture medium. (b) The fitting standard curve of lysosomal pH value in RAW-Blue. (c) The lysosomal pH value of RAW-Blue with or without cytokine IL-4 ($n = 16-18$).

8. *SI21 a), m1 and m2 are the untreated cells? And the particles are tested on which type of macrophages? please clarify.*

Re: Thanks for the reviewer's good suggestion. To avoid misleading, we have revised the legend in **Supplementary Figure 25**. In the reprogramming experiments, we incubated different groups of immune adjuvants with M2-like BMDMs, while M1- and M2-like BMDMs were the untreated cells and chosen as control groups.

Supplementary Fig. 25. Proinflammatory cytokines and acute systemic toxicity analysis. (a) ELISA assay of IL-12 secreted from BMDMs pretreated with IMDQ preparations (equivalent to 10 μ M IMDQ) for 24 hours.

9. *SI22) Please explain why the H&E staining shows good bio compatibility*

Re: Thanks for the reviewer's question. Compared with PBS group, major organs from mice treated with different formulations kept tissue structure intact without cell morphology change. Moreover, no necrosis and fibrosis were observed in the major organs. Thus, the H&E staining shows good biocompatibility of different nanoadjuvants. We have added the explanation in the main text.

Line 182-184: Hematoxylin and eosin (H&E) staining of major organs showed that pretreated mice kept tissue structure intact without cell morphology change in major organs, indicating a good biocompatibility of PGN_{4.9} nanoadjuvant (Supplementary Fig. 26).

10. *5A/B/C) Please also show the raw count rather than only percentages*

Re: Thanks for your kind advice. We have added the raw count of immune cells to compare the lymphocyte infiltration in tumours between different treatments as shown in **Supplementary Figure 37**.

Supplementary Fig. 37. Quantification of tumour infiltration for different immune cells. Cell numbers of (a) CD8⁺ cytotoxic T cells; (b) iNOS⁺ M1-like macrophages; (c) CD206⁺ M2-like macrophages; (d) CD4⁺ Foxp3⁺ regulatory T cells ($n = 20-25$). Statistical significance was analyzed by one-way ANOVA followed by Dunnett's multiple comparisons test

11. *SI32D*) Please discuss why there are equal amounts of M2 macrophages between the IMDQ, NPGN and PGN_{4,9}.

Re: Re: The reviewer raised a good point. Because of the high efficiency of TLR activation by IMDQ with a very low EC₅₀ (Rodell *et al*, *Nat Biomed Eng* **2018**, 2: 578-88), all the three IMDQ preparations down-regulated the M2-like tumour-associated macrophages.

However, unlike PBS, IMDQ and NPGN nanoadjuvant control groups, PGN_{4,9} nanoadjuvant could sustainably release large numbers of IMDQ drug for 48 h *in vivo*. Thus, PGN_{4,9}-treated mice presented a dramatically higher intratumoural ratios of M1-like TAMs to M2 subset, to which we paid more attention according to the reported literatures (Wang *et al*, *Sci Tran Med* **2021**, 13(615): eabb6981). As for the equal level of M2-like macrophage in different groups, it is probably due the feedback regulation after the macrophage repolarization (Chen *et al*, *Sig Transduct Target Ther* **2023**, 8, 207).

12. *5G*) Please explain how the CFSE dye correlates with killing.

Re: Thanks for the reviewer's question. The carboxyfluorescein succinimidyl ester (CFSE) method has been extensively utilized to quantify the OVA-specific CTL response (Luo *et al*, *Nat Nanotechnol* **2017**, 12, 648-654). Ovalbumin (OVA) was used as a model antigen. On the third day after the last treatment, immunized mice were intravenously injected with an equal number of OVA₂₅₇₋₂₆₄-presented splenocytes (CFSE^{high}) and naive splenocytes (CFSE^{low}). Mice with strong immunity can recognize OVA-presented splenocytes labelled with high level of CFSE (CFSE^{high}) for specific killing, while the naive splenocytes (CFSE^{low}) cannot be killed by OVA-specific CTL. Therefore, we investigated the changes in the proportion of two groups of splenocytes with high and low expression of CSFE to calculate the OVA-specific killing capacity.

13. *6I/J*) As the PDPA-DTX is not an established treatment that increases antigen presentation, please show this. It would also be interesting to see immune infiltration data for this experiment to determine how it worked.

Re: Thanks for the reviewer's question. The chemotherapeutic PDPA-DTX nanomedicine can directly kill tumour cells, triggering immunogenic cell death and tumour-associated antigen release, which has been proved in our previous studies (Du *et al*, *Adv Funct Mater* **2020**, 30(39): 2003757). Thus, the nanomedicine can in situ supply abundant tumour antigen and recruit macrophages to promote antigen cross-presentation in tumour tissues. When combined with PDPA-DTX nanomedicine, PGN_{4.9} further boosted intratumoural CD3⁺CD8⁺ T cells.

We also performed the new experiments to determine the immune cell infiltration in tumours treated with PDPA-DTX, or PDPA-DTX plus PGN_{4.9} nanoadjuvant. The immune infiltration data for the combination therapy have been included in **Supplementary methods** and **Supplementary Figure 44**.

Line 277-280: Combining with PDPA-DTX nanomedicine developed by our laboratory³⁴, PGN_{4.9} nanoadjuvant exhibited a greater tumour inhibition and prolonged survival with a reduced adjuvant dosage **through boosting intratumoural cytotoxic T cells**, as compared with nanoadjuvant treatment alone (Fig.6i, 6j; Supplementary Fig. 44).

Supplementary Fig. 44. (d) Representative flow cytometry plots and **(e)** quantification of CD3⁺CD8⁺ cytotoxic T lymphocytes in tumour tissues. **(f, g)** The percentage of CD3⁺CD4⁺ T lymphocytes from 4T1 tumour-bearing mice with different treatments (*n* = 4).

14. 6K) Why the sudden switch from 4T1 to MC38 in this figure? All previous experiments were only demonstrated for 4T1. Did they also work with MC38?

Re: Thanks for the reviewer's question. We have already confirmed the pH-gated imaging of intratumoural M2-like macrophages in MC38 tumour model as shown in **Figure 3g** and **Supplementary Figure 19**. In addition, we performed antigen presentation experiments in lymph node and specific T cell killing experiments from splenocytes in MC38 tumour model as shown in **Figure 5e-5j**. To investigate the immunological memory effect, the MC38 tumour model was selected since 6 out of 10 MC38 tumours were completely removed after PGN_{4.9} nanoadjuvant treatment, allowing for long-term rechallenge experiments.

15. 6L/M/N) The figures show a combination therapy using anti-PD1 and PGN_{4.9} IMDQ. However, only tumor growth kinetics and survival are shown. Please also show infiltration data to indicate how the combination therapy might work.

Re: We appreciate the reviewer's suggestion and have performed immune infiltration experiments in

the combination therapy. For the PGN_{4.9} group, 2.90-fold higher percentage of CD3⁺CD8⁺ leukocytes were observed as compared to the PBS group. The combination with anti-PD1 antibody further promoted the intratumoural infiltration of CD8⁺ T cells, which was 4.87-fold more than that in PBS treated tumours, with negligible influence on CD4⁺CD25⁺ regulatory T cells.

We included the infiltrating leukocytes studies in **Supplementary methods** and **Supplementary Figure 46**.

Line 291-294: The results showed that α-PD1 treatment alone led to no significant inhibition of tumour growth, whereas 70% of the mice immunized with α-PD1 and PGN_{4.9} nanoadjuvant exhibited no tumour progression within 70 days **by significantly promoting the intratumoural infiltration of CD8⁺ T cells** (Fig. 6l-6n; Supplementary Fig. 46).

Supplementary Fig. 46. (c) Representative flow cytometry plots and (d) quantification of CD3⁺CD8⁺ cytotoxic T lymphocyte in MC38 tumours. (e, f) The percentage of CD4⁺CD25⁺ regulatory T lymphocyte from mice with different treatments ($n = 5$).

16. Literature describes that activating existing M1 macrophages using imidazoquinoline TLR ligands can inhibit tumor growth. The authors should discuss why only targeting M2 macrophages would be more beneficial rather than targeting both type of macrophages.

Re: Thanks for the reviewer's good question. We agree that targeting M1- and M2-like TAMs is more beneficial for tumour inhibition. However, it is hard to specifically activate M1-like TAMs rather than M0- and M1-like macrophages in normal tissues. M2-like phenotype accounts for 60%-90% of tumour-associated macrophages in different tumour models. Targeting M2-like TAMs using PGN_{4.9} nanoadjuvant can not only repolarize M2-like phenotype to M1 ones, but also trigger the sustained release of IMDQ for persistent activation of the function of reprogrammed M1-like macrophages. Meanwhile, the released IMDQ probably plays a bystander effect for the activation of M1-like TAMs. Thus, PGN_{4.9} nanoadjuvant achieves targeting both type of macrophages for tumour inhibition. Combining the next question (Q17), we have added the discussion in the main text.

17. The work has not been put into perspective in the final section of the manuscript. It is important that the authors show the relevance in comparison to current literature on this topic.

Re: Thanks for the reviewer's suggestion. We have discussed the relevance in comparison to published literatures in the revised manuscript.

Line 313-336: Immunoagonists (e.g. R848 and CpG) in free form are easily distributed throughout the

body and cause severe systemic immunotoxicity, which hampers their clinical translation^{39,40}. A variety of nanoparticles have been developed to deliver immunoagonists and achieve the polarization of M2-like TAMs^{10,41,42}. Recent studies demonstrated that intravenous delivery might increase the possibility of effective co-localization of immunologic adjuvant with dying tumour cells, thus producing an in situ vaccination for superior immune responses⁴³. However, many such strategies based on the tissue targeting mechanism that could also activate M0- and M1-like macrophages in non-malignant organs including liver, spleen, lung and skins, raising biosafety concerns. Therefore, nanoadjuvant technology that specifically modulate M2-like TAMs would provide safe and effective formulations optimized for cancer immunotherapy.

Based on our finding that the lysosomal pH difference between M2-like TAMs ($pH_L \sim 4.4$) and M0- and M1-like macrophages ($pH_L \sim 5.2$), engineering of pH-responsive nanotechnology would be a promising approach to achieve targeted modulation of M2-like TAMs. So far, pH-responsive nanoparticles have been extensively reported to shuttle the therapeutic cargoes to solid tumours through pH-triggered nanocarrier disintegration or linkage cleavage upon pH changes within tumour microenvironment⁴⁴⁻⁴⁶. Although these approaches offer the targeted delivery of therapeutic cargoes to lysosomal compartments, achieving selective targeting to the highly lysosomal acidity of M2-like TAMs rather than that of other cells remains a significant challenge. In this article, our PGN nanoadjuvants were successfully designed and screened with several key features that enable the specific polarization of M2-like TAMs instead of other macrophages in normal tissues due to their AND-gated performance. Firstly, the PGN nanoadjuvants render a sharp pH response ($\Delta pH_{ON/OFF} \sim 0.2-0.3$), which is critical for the pH-gated activation in lysosomal compartment of M2-like TAMs rather than counterpart of other macrophages. Secondly, the pH_t tunability enables the successful screening of PGN_{4.9} nanoadjuvant for the specific targeting of lysosomal pH of M2-like TAMs, followed by enzymatic cleavage-mediated drug release to achieve logic-gated immunotherapies.

Reviewer #2 (Remarks to the Author): with expertise in tumor associated macrophages

Tang et al present pH-gated nanoparticles, termed PGN_{4.9}, that degrade in the lysosomal compartment of M2-oriented macrophages, but not M0/M1 or other normal cell types or cancer cells. The particles accumulate in the tumor microenvironment and, when loaded with the TLR7/8 ligand IMDQ, are able to skew the macrophage activation state to M1, with a higher antigen-presenting capacity and anti-tumor activities. Consequently, PGN_{4.9} collaborates with anti-PD1 to increase anti-tumor immunity. His is an interesting story, but some remarks remain:

We thank the reviewer's positive comments and provide a point-by-point response to the reviewer comments.

1. *I don't understand why PGN_{6.3} would accumulate less in the tumor. Accumulation is influenced by size, stability etc, so is this different for PGN_{6.3}?*

Re: According to our previous studies (Li *et al*, *Nanomedicine* **2019**, 17: 287-296), All three ICG-conjugated micelles had long blood circulation with half-lives of more than 10 h, while the pH_t of nanoparticles have significant effect on their blood AUC_{0-24 h}. Moreover, our group performed an in-depth research on the effect of hydrophobicity of micellar core on its pharmacokinetics behavior, finding that lower pH_t of nanoparticles would lead to longer circulation time *in vivo*, as shown in **Figure R1**. Therefore, the tumour accumulation in PGN_{4.9}- and PGN_{4.3}-treated groups was 1.93- and 2.06-fold of that in PGN_{6.3}-treated group due to the higher blood AUC for PGNs with lower pH_t. In other experiments from our group, the nanoparticles with lower pH_t would have a smaller critical micelle concentration (Wang *et al*, *Nature Materials* **2014**, 13: 204-212). Thus, the PGNs with lower pH_t would have a better serum stability, which allowing for the higher tumour accumulation.

Figure R1. Pharmacokinetics profiles of five ICG-conjugated nanoparticles with different pH_t (5.25-6.75) in tumour-free BALB/c mice ($n = 3$).

2. *The authors conclude that PGN_{4.9} does not get activated in normal tissues, based on observations in the liver and the spleen. However, the authors should look at organs where more M2-oriented macrophages are likely to reside, such as eg the lung and possibly also the skin.*

Re: We appreciate the reviewer's good suggestion and have carried out new experiments about the accumulation level and activation efficiency in other major organs, including lung, kidney, and skin. The results showed that the activation efficiency of PGN_{4.9} in different organs with M2-like macrophages were still significantly lower than that in tumour tissues. Therefore, we included the accumulation level and activation efficiency in different major organs in **Figure 3e** and **Supplementary methods**.

Figure 3. (e) Accumulation level and activation efficiency of PGN_{4.9} in dissected livers, spleens, lungs, kidneys, tumours and skin ($n = 4$ biologically independent mice, one-way ANOVA followed by Dunnett's multiple comparisons test).

3. Fig 3g shows a correlation between the ratiometric signal and the presence of CD206⁺ TAMs. However, tumors may also contain M1-like TAMs. How is the correlation with those TAMs? The authors should more directly demonstrate that PGN_{4.9} fluoresces only within the more M2-oriented TAMs

Re: Thanks for the good suggestion. The proportions of CD11b⁺F4/80⁺CD206⁻ macrophages were also measured by flow cytometry and the correlation analysis between M1-like macrophages content and ratiometric signal was analyzed by Graphpad Prism 8 software, as shown in **Figure R2**. The curve of ratiometric signal (Cy5/Cy7.5) versus the percentage of intratumoural M1-like macrophages exhibited a poor linear correlation ($R = 0.2621$, $P = 0.2642$). Combined with data in **Figure 3g**, the results further demonstrated the selectivity of PGN_{4.9} towards M2-like macrophages *in vivo*. Moreover, we also have verified that more than 80% of intratumoural PGN_{4.9} were internalized by M2-like macrophages, as shown in **Figure 3f**.

Figure R2. Linear correlation between ratiometric signals versus percentage of M2-like macrophages in different tumour models (4T1, MCF-7, CT26, MC38, PANC02) treated with PGN_{4.9}-Cy5/Cy7.5 binary nanoreporter.

4. Fig 4b also shows that PGN_{4.9} adjuvant (I presume that this is the PGN_{4.9} particle conjugated to IMDQ, although I don't find this very clear from the text) stimulates TLR a lot less than free IMDQ. Please explain.

Re: We appreciate the insightful questions by the reviewer. The PGN_{4.9} adjuvant has been replaced with PGN_{4.9} nanoadjuvant we have made an explanation in the main text. For the efficacy of TLR activation, the half maximal effective concentration (EC₅₀) of PGN_{4.9}-IMDQ was 3.84 µM, which was significantly higher than that of free IMDQ (EC₅₀ = 0.0124 µM). This big difference is probably due to several reasons. Firstly, the free IMDQ can diffuse freely into cells while PGN_{4.9} nanoadjuvant is internalized via the energy-dependent endocytosis, which is not as efficient as the former. Secondly, the internalized PGN_{4.9} nanoadjuvant needs to be cleaved by Cathepsin B for sustained drug release to take effect. Thirdly, all the EC₅₀ data were obtained by static cell culture that maintain the free IMDQ level across the experiments. However, the pharmacokinetics and biodistribution of PGN_{4.9} nanoadjuvant is totally different from the free IMDQ *in vivo*. PGN_{4.9} nanoadjuvant can efficiently accumulate into tumour tissues, sustainably release IMDQ to selectively activate the M2-like TAMs rather than M0- and M1-like tissue-resident macrophages.

In contrast, free IMDQ will distribute broadly into all the tissues after intravenous injection, and nonspecifically activate M2-like TAMs with low efficiency due to its poor tumour exposure, followed by significant off-target effect.

5. *The experimental conditions used in Figure 4 should be clearer represented on the figures or in the legend. If the authors use "PGN_{4,9}", what does that mean? I presume they mean M2 + PGN_{4,9}, but that's nowhere stated. Could also be M0 + PGN_{4,9}.*

Re: Thanks for the reviewer's good suggestion. To avoid misleading, we have revised the legend in **Figure 4**. In the reprogramming experiments, we incubated different groups of immune adjuvants with M2-like BMDMs for 24 hours, while M1- and M2-like BMDMs were the untreated cells and chosen as control groups.

6. *Along the same line, what are the conditions in Figure 4f? I guess "M2" means a treatment with PGN_{4,9}+OFF/ON fluorescent module, while "PGN_{4,9}" means PGN_{4,9}+IMDQ? I can only guess...*

Re: Thanks for the reviewer's good suggestion. We have revised all the legend and made the experimental condition clearer. In **Figure 4f**, M2 group means M2-like BMDMs treated with PGN_{4,9} nanoreporter, whereas M2+PGN_{4,9} group means M2-like BMDMs was firstly treated with PGN_{4,9} nanoadjuvant for repolarization, followed by the treatment with PGN_{4,9} nanoreporter for pH-gated reporting of lysosomal acidity.

7. *Fig 4m. Are the MHC-I/SIINFEKL complexes higher in the PGN_{4,9} condition because more SIINFEKL peptide is produced or because there is a higher overall surface expression of MHC-I molecules? The authors should discriminate between these possibilities by assessing the production of SIINFEKL in M2 versus M2+PGN_{4,9}*

Re: We acknowledge the reviewer's insightful question. Per the reviewer's suggestion, we performed the experiments to investigate the MHC-I molecule expression levels on the cell surface of M2-like BMDMs before and after PGN_{4,9} nanoadjuvant treatments. The results in **Figure 4k** and **Supplementary Figure 33** showed that PGN_{4,9} nanoadjuvant promoted both the generation of SIINFEKL peptide and expression of MHC-I molecule, leading to the enhanced antigen cross-presentation. We added the above results in **Supplementary methods** and **Supplementary Figure 33**.

Lines 222-224: Moreover, PGN_{4,9} nanoadjuvant caused a significantly higher MHC Class I molecules on the cell surface of M2-like BMDMs after repolarization to M1-like phenotype (Supplementary Fig. 33).

Supplementary Fig. 33. The expression of MHC-I molecules on BMDMs. Mean fluorescence intensity of H-2K^d MHC class I molecules expressed on cell membranes after treatment with different IMDQ formulations (equivalent to 10 μM IMDQ) for 24 h (n = 3).

8. *The authors investigated the biodistribution of the particles in the tumor, but not the tumor-draining lymph node. In Fig 5, however, they suddenly look at the tdLN macrophages instead of the TAMs. What happens to the TAMs in terms of antigen processing and presentation?*

Re: Per the reviewer's advice, we investigated the antigen peptide presentation of tumour-associated macrophages in tumour tissues of MC38.OVA tumour-bearing mice by flow cytometry. As shown in **Supplementary Figure 38**, we found that the intravenous injection of PGN_{4.9} nanoadjuvant increased 9.8-fold MHC class I-associated SIINFEKL displayed on the cell membrane of TAMs with the capacity to present antigen. Therefore, PGN_{4.9} nanoadjuvant promoted effective antigen cross-presentation both in tumour-associated macrophages or tLN macrophages. We added the above results in **Supplementary methods** and **Supplementary Figure 38**.

Lines 248-250: Meanwhile, PGN_{4.9} nanoadjuvant also induced an increased presentation of cell-surface SIINFEKL peptide on macrophages from tumour-draining lymph nodes and **intratumoural TAMs** of immunized mice (Fig. 5f; **Supplementary Fig. 38d-f**).

Supplementary Fig. 38. (e) Representative flow cytometry plots of SIINFEKL⁺ cells in CD11b⁺F4/80⁺ macrophages from MC38.OVA tumours. **(f)** Flow cytometry of OVA₂₅₇₋₂₆₄-positive macrophages in TAMs.

In addition, we performed the immunofluorescence staining of F4/80 antibody (macrophage biomarker) in lymph node sections. Results confirmed that PGNs had a good colocalization with macrophages within lymph nodes. We added the above results in **Supplementary methods** and **Supplementary Figure 21**.

Lines 160-162: Besides, PGN_{4.9} also had highest fluorescence signals in tumour-draining lymph nodes as compared with other groups, which were co-localized well with macrophages (Supplementary Fig. 21).

Supplementary Fig. 21. Biodistribution of PGNs in lymph nodes from 4T1 tumour-bearing mice. (a) Nanoparticle level in excised lymph nodes at 24 hours after administration of ICG-conjugated PGNs (20 mg kg⁻¹). **(b)** Mean fluorescence intensity of ICG signals in lymph nodes (*n* = 4-5). **(c)** Whole-mount images of lymph node slices at 24 h post-administration of Cy5-labelled PGN_{4.9} and control group. Scale bar, 50 µm. The macrophages were stained with anti-F4/80 antibody.

9. Fig 5h. Why does NPGN result in a higher killing capacity than IMDQ?

Re: Although free IMDQ has the highest activation efficiency to TLR, it washed out very rapidly *in vivo*, resulting in a low tumour accumulation and thus poor immune response in MC38.OVA tumours after intravenous injection. However, NPGN nanoadjuvant was efficiently accumulated into tumours through enhanced permeability and retention (EPR) effect due to their nanostructure. After tumour accumulation, NPGN nanoadjuvant inevitably released a small amount of free IMDQ in tumour tissues for a persistent activation of immune response, producing a certain specific killing (66.3%). In contrast, PGN_{4.9} can effectively deliver large numbers of IMDQ into TAMs, achieving the most efficient specific killing capability (92.3%).

10. Fig 6h. This means that these TAMs contribute to anti-tumor immunity. Do these TAMs kill cancer cells? In addition, to what extent are CD8⁺ T cells important? This can be shown by *in vivo* depletion of these cells.

Re: Thanks for the reviewers' good question. It has been demonstrated that macrophages have the potential to kill tumour cells, mediate antibody-dependent cellular cytotoxicity and phagocytosis, elicit vascular damage and tumour necrosis. Therefore, macrophage reprogramming induces macrophage-mediated killing of cancer cells (Mantovani *et al*, *Nat Rev Drug Discov* **2022**, 21, 799-820; Singhal *et al*, *Sci Transl Med* **2019**, 11, eaat1500).

We also investigated the significance of CD8⁺ T cells in PGN_{4.9} nanoadjuvant mediated tumour attenuation, and the results are shown in **Supplementary methods** and **Supplementary Figure 43**. Anti-CD8 antibody was injected intraperitoneally to deplete CD8⁺ T cells in orthotopic 4T1 tumour-bearing mice. The effects of PGN_{4.9} nanoadjuvant on tumour inhibition was mitigated significantly in T lymphocytes-depleted mice. Thus, CD8⁺ T cells and macrophages are both important to PGN_{4.9} adjuvant immunotherapy.

Line 272-274: The effects of PGN_{4.9} nanoadjuvant on tumour inhibition was significantly mitigated by depletion of 76.8% macrophages in tumour-bearing mice (Fig. 6h). T cell depletion also led to tumour relapse in mice treated with PGN_{4.9} nanoadjuvant (Supplementary Fig. 43).

Supplementary Fig. 43. In vivo depletion of T cells in 4T1 breast tumour model. (a) Schematic illustration for mechanism study of immunotherapy. **(b)** Representative flow cytometry plots and repopulation of CD3⁺CD8⁺ cytotoxic T lymphocytes in peripheral blood of mice pretreated with PBS or anti-mouse CD8 antibody three times ($n = 5$). **(c)** Tumour immunotherapy after macrophage depletion in 4T1 tumour-bearing mice ($n = 7-9$). **(d)** Body-weight changes of mice in various groups.

11. Fig 6k. The control group says "PBS", while these are simply control C57BL/6 mice.

Re: Per the reviewer's suggestion, we have made correction in **Figure 6k** according to your comments.

The caption of Figure 6k: On day 73 and day 120, PGN_{4,9} nanoadjuvant treated mice in MC38 colorectal tumour model were rechallenged with MC38 cells (2×10^6 cells/mouse). For the control groups, naïve C57BL/6 mice were subcutaneously injected with the same number of MC38 cells ($n = 9$ mice at day 73 and $n = 6$ mice at day 120; two-tailed unpaired Student's t-test).

Reviewer #3 (Remarks to the Author): with expertise in nanoparticles

In this manuscript, a pH-gated nanoparticles library was established for selectively target the highly lysosomal acidity of M2-like macrophages. The PGN_{4.9} nanoadjuvant with AND-gate capacity can selectively fine-tune lysosomal proteolysis of M2-like TAMs to facilitate antigen cross-presentation and provoke adaptive tumour immunity. The study is interesting, but some results are not very persuasive. Based on these points, I suggest that this manuscript can be accepted for publication after addressing following issues.

We appreciate the Reviewer's encouraging comments on the novelty and impact of the PGN_{4.9} nanoadjuvant nanotechnology. We have revised the manuscript according to your advices and other referee's comments.

1. *Line 437, the authors adopted fluorescence intensity to evaluate the PGN accumulation and activation efficiency. This method of quantitative analysis is not accurate. The PGN with the fluorescent label attached may have residual fluorescent label, or the fluorescent label may be free from the polymer. To assess the accumulation or metabolism of PGN, a more accurate method is still chromatography or chromatography-tandem mass spectrometry.*

Re: Thanks for the reviewer's kind suggestion. In our previous studies, we have evaluated the pharmacokinetics and tumour accumulation of PGNs using radio-labelled analysis and fluorimetry analysis. For the fluorimetry, multiple fluorescent dyes (e.g., TMR, Cy5 and ICG) were conjugated to the polymers through amide bond, which is very stable in physiological environment as shown in **Supplementary Figure 6**. The free dye was completely removed through ultracentrifugation. All these dye-conjugated polymeric nanoparticles present an identical pharmacokinetics and biodistribution profiles. Moreover, we compared the pharmacokinetics and biodistribution profiles of PGN nanoparticles between radio-labelled analysis and fluorimetry analysis (Wang Y, *et al*, *Nat Mater* **2014**, 13(2): 204-212). The results also showed a very similar PK and PD behaviors. In addition, it is very difficult to quantify the polymer concentration using chromatography or chromatography-tandem mass spectrometry. Taken together, we utilized fluorescence labeling method to evaluate the PGN accumulation and activation efficiency.

2. *After PGN_{4.9} was injected into animals through the tail vein, how does it target the M2-like TAM in tumors? What is the mechanism of this targeting function? How was it considered when designing the synthesis?*

Re: After intravenous injection, PGN_{4.9} was passively accumulated into tumour tissues through enhanced permeability and retention (EPR) effect, and internalized by different kinds of cell types in tumour microenvironment (TME). Among the cells in the TME, tumour-associated macrophages that account for high percentage (up to 60-90%) and localize in the perivascular area have a high chance to take up a large number of nanoparticles. Due to the highly lysosomal acidity of M2-like macrophages, PGN_{4.9} could selectively be activated in lysosomal pH (pH_L ~ 4.4) of M2-like macrophages, while kept silent in lysosomal pH (pH_L ~ 5.2) of other cells. Therefore, PGN nanotechnology achieved targeting M2-like TAMs activation.

Furthermore, PGN_{4.9} was colocalized well with lysosomes, which are the location of cathepsin family and TLR7/8. So, we conjugated a TLR-7/8 agonist to the hydrophobic block of pH-gated nanoparticle through Gly-Phe-Leu-Gly (GFLG) linkage. The tetrapeptide linker can be stable in delivery process but be cleaved by the lysosomal cathepsin B, thereby releasing free IMDQ and stimulating TLR7/8 in endo-lysosomes. Thus, the PGNs can efficiently accumulated in tumour tissues, be internalized by M2-like TAMs, then be dissociated and cleaved in lysosomes of M2-like TAMs, and finally release IMDQ for specific TLR activation and polarization of M2-like TAMs to M1 phenotype.

3. *By using what kind of method to verify the lysosomal pH value in BMDMs to obtain the fitting standard curve? Can you provide more experiment protocols?*

Re: We utilized ratiometric pH quantification protocol to determine the lysosomal pH of different living cell types. Firstly, we present the incorporation of the pH-sensitive fluorophore Oregon Green-dextran and the pH-insensitive fluorophore Rhodamine B-dextran into a fluorescent cocktail. The Oregon Green ($\lambda_{\text{ex}}/\lambda_{\text{em}} = 488/525$) shows a pH-sensitive enhancement of fluorescence signals as pH increase from 4.0 to 6.0, while Rhodamine B ($\lambda_{\text{ex}}/\lambda_{\text{em}} = 561/595$) presents constant fluorescence readouts regardless of pH fluctuation. Therefore, the ratiometric signal of Oregon Green versus Rhodamine B was obtained at each pH value that is independent of dye concentration. The ratio as a function of pH was plotted to obtain the fitting standard curve for pH calculation. We have included measurement protocols of lysosomal pH in methods.

Line 380-388: Cell lines were pulsed with mixed fluorescent dextran for 6 h, chased for 12 h in fresh medium, and washed with PBS to calculate the fluorescence intensity of lysosomes by a confocal microscope (A1R-Storm, Nikon) under 60 \times oil objective lens. Subsequently, a corresponding calibration was performed for each lysosome. The cells were incubated with nigericin (10 μM) in high- K^+ buffers from pH 4.0 to 6.0 for 5 min equilibrium on the ice. The fluorescence intensity was measured by the confocal microscope. The Hoechst 33342 (Invitrogen), Oregon Green, and Rhodamine B were excited at 405 nm, 488 nm, and 561 nm, respectively. The ratiometric images were processed by NIS-Elements viewer software and the resulting quantification was calculated by ImageJ software (NIH), which was plotted as a function of pH value and fitted to a Boltzmann sigmoid to measure the lysosomal pH in various cell lines.

4. *The manuscript contains too many abbreviations, which makes the reading process not very smooth. The meaning of the acronym is often forgotten. If the phrase or word composition is not complex, try to use the original expression to make the article reading more smoothly. If a phrase occurs only once or twice, it is not necessary to assign a separate abbreviation. For example, TME stands for tumor microenvironment, APCs for antigen-presenting cells, and GPC for chromatography. It is not necessary to create an abbreviation for these phrases or words.*

Re: Per the reviewer's suggestion, we have corrected the original expression and made some changes in the revised manuscript. These changes will not influence the content and framework of the paper.

5. *In supplementary Fig. 5, in the plot, the author should elaborate what is the F instead for. End explain the Two coordinate system in accuracy, avoiding ambiguous understanding. What F/F_{min} instead for?*

Re: Thanks for the suggestion. F/F_{min} represents the pH-dependent fluorescence signal amplification relative to micelle states. We have added the definition of F and F_{min} in Supplementary Fig. 5.

The caption of Supplementary Fig. 5: F is the fluorescence intensity of PGNs at any given pH value, and F_{min} represents the minimal fluorescence intensity at OFF state, respectively.

6. *In supplementary Fig. 5, the fluorescence intensity of Cy3.5 is stable, while the BDP is fluctuate dramatically. Why?*

Re: In our ultra-pH-sensitive nanotechnology, the tertiary amine groups incorporated into the copolymers act as ionizable units to impart ultra-pH-sensitivity. At $\text{pH} > \text{pH}_i$, the hydrophobic segments of copolymer self-assemble into the micelle cores, leading to fluorescence quenching of dye signals by homo- and hetero-FRET (Ma, *et al*, *JACS*, **2014**, 136, 11085-11092). Upon $\text{pH} < \text{pH}_i$, protonation of the hydrophilic segments results in micelle dissociation into unimers with fluorescent signal amplification owing to the increased distance between dyes. In our previous studies, more than twenty dye molecules (e.g. Coumarins, BDP, TMR, Cyanine dyes) were conjugated to the hydrophobic segments of copolymers. Among these dyes, the Cy3.5-conjugated polymeric nanoprobe only presents 2-fold fluorescence increase between dissociation and micelle states. In contrast, the BDP-conjugated nanoprobe shows more than

100-fold fluorescence amplification upon pH-triggered micelle dissociation from micelle state. Thus, to prepare the binary ratiometric nanoreporter, the BDP-conjugated and Cy3.5-conjugated copolymers were chosen as OFF-ON and always-ON modules, respectively. To achieve the always-ON signal, a low molar fraction of Cy3.5-conjugated copolymers (i.e. 40%) in the binary ratiometric nanoreporter was used to abolish the HomoFRET-induced fluorescence quenching in the micelle state. As a result, the Cy3.5 module presents a constant fluorescence signal across a broad pH range (4.0-7.4) regardless of dissociation and micelle states. In contrast, the OFF-ON (BDP) module renders more than 60-fold signal amplification when pH drops below the pH_t of each nanoreporter. The design and in-depth mechanism of binary ratiometric nanoreporter was reported in our previous work (Yin *et al*, *Nat Commun* **2021**,12, 2385).

7. *In Line 422, “tumors were frozen and cut into several adjacent slices with 10 μ m thickness”. Do you mean “Tumor tissues were cryo-sectioned and sliced into 10-micron slices.” The type of instrument that used, and the freezing temperature should be described.*

Re: Thanks for the reviewer’s good suggestion. Information on instrument type and freezing temperature has been supplemented in the revised manuscript.

Line 472-473: The excised 4T1-GFP tumours were cryosectioned and sliced into adjacent 10 μ m slices at -20 °C by Cryostat (Leica CM1950).

8. *The design, grouping and experimental cycle of animal experiments are not well described. How many animals in each group need to be maintained for 150 days? During this period, in addition to the change and control of tumor volume, how can animal welfare be guaranteed? Is there a humanitarian end? Are other indicators of animals tracked and monitored? Are there any abnormalities in behavior, physiological and biochemical indicators?*

Re: We acknowledge the reviewer’s kind suggestion and added the supplementary information in the caption of **Figure 6k**. Based on literature research (Rodell *et al*, *Nat Biomed Eng* **2018**, 2: 578-88; Donaldson *et al*, *J ImmunoTher Cancer* **2017**, 5: 69), we have designed the following experimental plan. On day 73 and day 120, tumour-free mice from MC38 tumour-bearing mice treated with PGN_{4.9} nanoadjuvant were rechallenged with MC38 tumour cells. For the control groups, naïve C57BL/6 mice were subcutaneously injected with the same number of MC38 cells ($n = 9$ mice at day 73 and $n = 6$ mice at day 120).

During the experiment, we regularly monitored the behaviour and physiological welfare indicators in the tumour-bearing mice. No abnormalities were found in terms of the behaviour, physiological, and biochemical indicators of the mice during experiments. All care and handling of animals were performed with the approval of the Ethics Committee of Peking University (Accreditation number: LA 2019039). When the tumour volume reached 1500 mm³, MC38 tumour-bearing mice were euthanized to prevent further distress.

The caption of Figure 6k: On day 73 and day 120, PGN_{4.9} nanoadjuvant treated mice in MC38 colorectal tumour model were rechallenged with MC38 cells (2×10^6 cells/mouse). For the control groups, naïve C57BL/6 mice were subcutaneously injected with the same number of MC38 cells ($n = 9$ mice at day 73 and $n = 6$ mice at day 120; two-tailed unpaired Student’s t-test).

9. *A thoroughly check should be made to improve the English and correct the grammar and typo.*

Re: Per the reviewer’s suggestion, we have gone through the manuscript carefully to check and revise typos/errors.

Reviewer #4 (Remarks to the Author): with expertise in nanomedicine, cancer immunology

This manuscript by Tang et al, presents an innovative approach consisting in the use of pH-gated nanoparticles with ability to deliver a drug (imiquimod) and regulate lysosomal function in tumor associated macrophages. The hypothesis, performance of experiments and results are appropriated, followed by a good presentation (both writing and figures). As the key element of this research, the development of pH-gated nanoparticles and their performance is very interesting and providing very satisfactory results, thus I recommend this manuscript for publication with minor delay after addressing the following minor concerns.

We appreciate the Reviewer's encouraging comments on the novelty and impact of the PGN nanotechnology. We provide a point-by-point response to the reviewer comments.

1. Supp fig 6 legend is wrong. b) and C) have to be changed.

Re: We appreciate the reviewer's advice. We have made correction in **Supplementary Figure 6** according to your comments.

2. Supp fig 13b) requires more details for the performance of the readout. 13c) legend requires description of organs collected. It is not clear if the 4T1 model is s.c. or orthotopic (probably s.c.). These details should be clearly mentioned in the figure legend.

Re: Per the reviewer's advice, we have added the legend of major organs in **Supplementary Figure 15c**. The 4T1 model is orthotopic model and related information was included in the figure caption.

Supplementary Fig. 15. In vivo long-term monitoring of PGNs distribution in orthotopic 4T1 breast tumour xenografts. (a) Representative fluorescent images of 4T1 tumour-bearing mice at selected time-points after intravenous injection of three ICG-conjugated PGNs (20 mg kg⁻¹) by IVIS Spectrum imaging system (n = 4). (b) Time-dependent fluorescence signals were quantified after injection of PGNs. (c) Nanoparticle level in excised tumour and major organs at 24 hours after administration of ICG-conjugated PGNs (20 mg kg⁻¹). (d) Mean fluorescence intensity of ICG signals in excised tumour and different organs.

3. *Dramatic decrease of signal is too exaggerated for figure 3.h and j. or supp. fig. 17. Some signal is still visible in tumors after depleting macrophages. I miss the analysis of macrophage depletion? which percentage of macrophages was depleted? only M2 or also M1 were depleted? For how long?*

Re: We acknowledge the reviewer's kind suggestion. We have revised this sentence according to the statistical analysis. Intravenous injection of clophosome every five days for three times could deplete macrophages, including both M1- and M2-like phenotypes (Zeisberger *et al*, *Br J Cancer* **2006**, 95: 272-281). Flow cytometry and immunofluorescence staining showed that tumour-associated macrophages can be cleared to less than 10% upon clophosome, but they cannot be completely depleted. Therefore, clophosome significantly decreased signal activation of PGN_{4.9} as compared to control groups.

Line 158-160: Moreover, depletion of 58.8% M2-like TAMs with clophosome caused a significantly decreased signal activation of PGN_{4.9}, further elucidated the selective targeting of PGN_{4.9} to M2-like TAMs (Fig. 3h-j; Supplementary Fig. 20).

4. *In supp. fig. 21 e-h, injected dose of IMDQ and others is missing.*

Per the reviewer's suggestion, we have supplemented the intravenous dose of different immune adjuvants (equivalent to 2 mg kg⁻¹ IMDQ) in **Supplementary Fig. 25**.

The caption of Supplementary Fig. 25: (e, f) A typical proinflammatory cytokine IP-10 was measured from serum of C57BL/6 mice injected intravenously with free IMDQ, PGN_{6.3} and PGN_{4.9} nanoadjuvants for 3 h and 48 h (equivalent to 2 mg kg⁻¹ IMDQ).

5. *In supp. fig. 27, indication of experimental doses and times would be appreciated.*

Re: Per the reviewer's advice, we have added the experimental dose of different IMDQ groups (equivalent to 10 μM IMDQ) and incubation time (24 hours) in vitro reprogramming experiment.

The caption of Supplementary Fig. 31: (b) Mean fluorescence intensity of DQ-OVA in BMDMs upon treatment with different IMDQ groups (equivalent to 10 μM IMDQ) for 24 h by flow cytometry.

6. *In supp. fig. 33, dose of each treatment would be appreciated.*

Re: Per the reviewer's advice, the administered dose has been included in the revised manuscript.

The caption of Supplementary Fig. 38: (b) FACS gating strategy for stratification of macrophages in tumour-draining lymph nodes from mice treated with different IMDQ formulations (equivalent to 2 mg kg⁻¹ IMDQ).

7. *In fig. 6, doses of treatments would be appreciated.*

Re: Per the reviewer's advice, we have added experimental details to evaluate the in vivo therapeutic efficacy of PGN_{4.9} nanoadjuvant in the caption of **Figure 6**.

The caption of Figure 6: (b) Individual tumour growth kinetics and **(c)** average tumour growth curves of 4T1-luc tumour-bearing mice treated with PBS, free IMDQ, NPGN and PGN_{4.9} nanoadjuvants (equivalent to 2 mg kg⁻¹ IMDQ, *n* = 7 biologically independent mice). **(k)** On day 73 and day 120, PGN_{4.9} nanoadjuvant treated mice in MC38 colorectal tumour model were rechallenged with MC38 cells (2 × 10⁶ cells/mouse). For the control groups, naïve C57BL/6 mice were subcutaneously injected with the same number of MC38 cells (*n* = 9 mice at day 73 and *n* = 6 mice at day 120; two-tailed unpaired Student's *t*-test). **(m)** Tumour progression and **(n)** survival curves of MC38 tumour-bearing mice treated with PBS, α-PD1 (100 μg per dose), PGN_{4.9} nanoadjuvant (equivalent to 2 mg kg⁻¹ IMDQ) and combined administration (*n* = 10-11 biologically independent mice; two-tailed unpaired Student's *t*-test).

8. *In supp. fig. 36, time of evaluation must be indicated.*

Re: Per the reviewer's kind suggestion, we have complemented the experimental schedule of anti-metastasis study.

The caption of Supplementary Fig. 41: (a) Bioluminescence images of ex vivo lung metastases in 4T1-Luc tumour model upon various IMDQ treatments visualized on day 28 by IVIS Spectrum imaging system ($n = 7$).

9. In line 254, the sentence "The effects of PGN_{4.9} on tumour inhibition was mitigated obviously in macrophages-depleted mice" must be reformulated, because the "mitigation" is not so obvious indicating that other cell types are also important. It could be also mentioned that macrophage depletion is not complete with the CSF1R antibody.

Re: Thanks for the reviewer's suggestion. We have revised the manuscript according to the statistical analysis. Moreover, we have investigated the significance of CD8⁺ T cells in tumour attenuation mediated by PGN_{4.9} nanoadjuvant, and the results are shown in **Supplementary methods** and **Supplementary Figure 43**. Anti-CD8 antibody was injected intraperitoneally to deplete CD8⁺ T cells in orthotopic 4T1 tumour-bearing mice. The effects of PGN_{4.9} nanoadjuvant on tumour inhibition was significantly mitigated in T lymphocytes-depleted mice. Thus, CD8⁺ T cells and macrophages are both important to PGN_{4.9} adjuvant immunotherapy.

Line 272-274: The effects of PGN_{4.9} nanoadjuvant on tumour inhibition was significantly mitigated by depletion of 76.8% macrophages in tumour-bearing mice (Fig. 6h). In addition, T cell depletion also led to tumour relapse in mice treated with PGN_{4.9} nanoadjuvant (Supplementary Fig. 43).

Supplementary Fig. 43. In vivo depletion of T cells in 4T1 breast tumour model. (a) Schematic illustration for mechanism study of immunotherapy. **(b)** Representative flow cytometry plots and repopulation of CD3⁺CD8⁺ cytotoxic T lymphocytes in peripheral blood of mice pretreated with PBS or anti-mouse CD8 antibody three times ($n = 5$). **(c)** Tumour immunotherapy after macrophage depletion in 4T1 tumour-bearing mice ($n = 7-9$). **(d)** Body-weight changes of mice in various groups.

10. For supp. fig. 38 and fig. 6i, the timing and doses of each therapeutic approach are not clear. These should be indicated.

Re: Per the reviewer's advice, we have added the experimental protocol of therapeutic strategy to clarify the timing and doses in tumour combination therapy.

The caption of Figure 6: (i) On day -7, BALB/c mice were inoculated subcutaneously with 4T1 cells. Tumour progression and (j) survival curves of 4T1 tumour-bearing mice treated with PGN_{4.9} nanoadjuvant

(equivalent to 1 mg kg⁻¹ IMDQ) and PDPA-DTX nanomedicine (equivalent to 3 mg kg⁻¹ DTX), alone or in combination on day 0, 4, 8 and 12 ($n = 9$ biologically independent mice).

11. Discussion must be significantly improved according with the following comments. a) In the first paragraph of discussion more references are needed. b) More details and references about the nanotechnology would be appreciated (other pH sensitive NPs). c) drug delivery approach to target macrophages in tumors require appropriate discussion.

Re: Thanks for the reviewer's suggestion. The literatures about tumour-associated macrophages and nanotechnology have been cited in the revised manuscript.

Line 297-304: Tumour-associated macrophages represent a heterogeneous population with distinct functions across many cancer types³. It is necessary to understand the dynamic interactions between TAMs and other infiltrated immune cells at the different stages of tumour progression³⁵. Currently, some anti-TAMs drugs are under preclinical and clinical evaluation, consisting of three main strategies⁴: (i) inhibition of TAMs recruitment³⁶, (ii) TAMs depletion³⁷, and (iii) re-education of M2-like TAMs³⁸. However, previous studies have demonstrated that the inhibition of TAMs recruitment and survival might not suffice to stimulate durable anti-tumour response¹⁰, whereas the re-education strategy represents a more effective choice to not only ameliorate the immunosuppressive functions but also to potentiate antigen cross-presentation.

Line 313-336: Immunoagonists (e.g. R848 and CpG) in free form are easily distributed throughout the body and cause severe systemic immunotoxicity, which hampers their clinical translation^{39,40}. A variety of nanoparticles have been developed to deliver immunoagonists and achieve the polarization of M2-like TAMs^{10,41,42}. Recent studies demonstrated that intravenous delivery might increase the possibility of effective co-localization of immunologic adjuvant with dying tumour cells, thus producing an in situ vaccination for superior immune responses⁴³. However, many such strategies based on the tissue targeting mechanism that could also activate M0- and M1-like macrophages in non-malignant organs including liver, spleen, lung and skins, raising biosafety concerns. Therefore, nanoadjuvant technology that specifically modulate M2-like TAMs would provide safe and effective formulations optimized for cancer immunotherapy.

Based on our finding that the lysosomal pH difference between M2-like TAMs (pH_L ~ 4.4) and M0- and M1-like macrophages (pH_L ~ 5.2), engineering of pH-responsive nanotechnology would be a promising approach to achieve targeted modulation of M2-like TAMs. So far, pH-responsive nanoparticles have been extensively reported to shuttle the therapeutic cargoes to solid tumours through pH-triggered nanocarrier disintegration or linkage cleavage upon pH changes within tumour microenvironment⁴⁴⁻⁴⁶. Although these approaches offer the targeted delivery of therapeutic cargoes to lysosomal compartments, achieving selective targeting to the highly lysosomal acidity of M2-like TAMs rather than that of other cells remains a significant challenge. In this article, our PGN nanoadjuvants were successfully designed and screened with several key features that enable the specific polarization of M2-like TAMs instead of other macrophages in normal tissues due to their AND-gated performance. Firstly, the PGN nanoadjuvants render a sharp pH response ($\Delta\text{pH}_{\text{ON/OFF}} \sim 0.2-0.3$), which is critical for the pH-gated activation in lysosomal compartment of M2-like TAMs rather than counterpart of other macrophages. Secondly, the pH_t tunability enables the successful screening of PGN_{4.9} nanoadjuvant for the specific targeting of lysosomal pH of M2-like TAMs, followed by enzymatic cleavage-mediated drug release to achieve logic-gated immunotherapies.

REVIEWERS' COMMENTS

Reviewer #1 (Remarks to the Author):

All my comments have been addressed.

Reviewer #2 (Remarks to the Author):

The authors successfully addressed my concerns

Reviewer #3 (Remarks to the Author):

In this manuscript, the authors have developed a novel drug delivery strategy using pH-gated nanoparticles (PGN4.9) for the delivery of a drug ((imiquimod, IMDQ), and have demonstrated that this nanocarrier can induce the differentiation of M2 macrophages into M1 macrophages in the tumor microenvironment. PGN4.9-IMDQ itself exhibits reduced tumor growth and can be used in combination with other immunotherapies to further decrease tumor growth. The scientific hypothesis, experimental design, and results of this study have been adequately and appropriately validated. The revised manuscript presents comprehensive and compelling evidence.

According to the author's reply, in this manuscript, the tumor targeting strategy adopted by the authors is based on the EPR (enhanced permeability and retention) effect. There is one more thing should pay attention to in future research. While the EPR effect is widely held to improve delivery of nanodrugs to tumors, it in fact offers less than a 2-fold increase in nanodrug delivery compared with critical normal organs, resulting in drug concentrations that are not sufficient for curing most cancers. Whether the enhanced permeability and retention effects are sufficient to cure cancer remains a matter of debate.

Overall, the paper has addressed the scientific hypothesis, experimental design, and results effectively. The authors have made significant contributions to the field of drug delivery and immunotherapy. However, the reliance on the EPR effect for tumor targeting raises concerns about the efficacy of the nanocarrier. Further investigation into alternative tumor

targeting mechanisms would enhance the impact and translational potential of this research. But this does not affect the innovation and contribution of this study. Therefore, I recommend this manuscript for publication.

Reviewer #4 (Remarks to the Author):

I congratulate the authors for the revision of the manuscript, appropriately addressing reviewers' concerns. Thus, I recommend the manuscript for publication without further delay.